# Proteins that carry dual targeting signals can act as tethers between peroxisomes and partner organelles

**Elena Bittner**[1☯], **Thorsten Stehlik**[1☯], **Jason Lam**[2☯], **Lazar Dimitrov**[2], **Thomas Heimerl**[3,4],
**Isabelle Schöck**[1], **Jannik Harberding**[1], **Anita Dornes**[3,4], **Nikola Heymons**[1], **Gert Bange**[3,4],
**Maya Schuldiner**[5], **Einat Zalckvar**[5], **Michael Bölker**[1,4], **Randy Schekman**[2]*,
**Johannes Freitag**[1,2]*

1 Department of Biology, Philipps-University Marburg, Marburg, Germany, 2 Department of Molecular and Cell Biology and Howard Hughes Medical Institute, University of California, Berkeley, California, United States of America, 3 Department of Chemistry, Philipps-University Marburg, Marburg, Germany, 4 Center for Synthetic Microbiology, Philipps-University Marburg, Marburg, Germany, 5 Department of Molecular Genetics, Weizmann Institute of Science, Rehovot, Israel

☯ These authors contributed equally to this work.
* schekman@berkeley.edu (RS); johannes.freitag@biologie.uni-marburg.demailto: (JF)

**Data Availability Statement:** All relevant data are within the paper and its Supporting Information files.

## Abstract

Peroxisomes are organelles with crucial functions in oxidative metabolism. To correctly target to peroxisomes, proteins require specialized targeting signals. A mystery in the field is the sorting of proteins that carry a targeting signal for peroxisomes and as well as for other organelles, such as mitochondria or the endoplasmic reticulum (ER). Exploring several of these proteins in fungal model systems, we observed that they can act as tethers bridging organelles together to create contact sites. We show that in *Saccharomyces cerevisiae* this mode of tethering involves the peroxisome import machinery, the ER–mitochondria encounter structure (ERMES) at mitochondria and the guided entry of tail-anchored proteins (GET) pathway at the ER. Our findings introduce a previously unexplored concept of how dual affinity proteins can regulate organelle attachment and communication.

## Introduction

Metabolic pathways in eukaryotic cells are often compartmentalized inside membrane-enclosed organelles such as mitochondria and peroxisomes [1]. Proteins synthesized in the cytosol must target and translocate into their organelles of residence correctly, efficiently, and in a regulated manner [2–4]. Most mitochondrial proteins are imported from the cytosol in a conformationally flexible state via evolutionary conserved protein complexes called translocase of the outer membrane (TOM) and translocase of the inner mitochondrial membrane (TIM). Many proteins destined for the mitochondrial matrix or inner membrane contain N-terminal targeting signals that are cleaved by proteases upon import [3].

Translocation of peroxisomal matrix proteins requires distinct targeting signals termed peroxisome targeting signal type 1 and 2 (PTS1 and PTS2, respectively). PTS1 motifs are

**Funding:** TS received funding from a fellowship from DAAD (https://www.daad.de/de/). GB thanks the European Research Council (ERC) for support through the project "KIWIsome" (Grant agreement number: 101019765). The project in the Schuldiner laboratory was supported by funding from ERC under the European Union's Horizon 2020 research and innovation programme (Grant agreement No. 864068). The robotic system of the Schuldiner laboratory was purchased through the support of the Blythe Brenden-Mann Foundation. MS is an Incumbent of the Dr. Gilbert Omenn and Martha Darling Professorial Chair in Molecular Genetics. RS is an investigator of the Howard Hughes Medical Institute. JF was supported by a fellowship from Leopoldina and by the German research foundation (FR 3586/2-1). The funders had no role in study design, data collection and analysis, decision to publish, or preparation of the manuscript.

**Competing interests:** The authors have declared that no competing interests exist.

**Abbreviations:** AID, auxin-inducible degron; CSM, complementary supplement mixture; ER, endoplasmic reticulum; ERMES, endoplasmic reticulum–mitochondria encounter structure; IMP, inner membrane peptidase; MSG, monosodium glutamate; MTS, mitochondrial targeting signal; PNS, post-nuclear supernatant; PTS, peroxisome targeting signal; RFP, red fluorescent protein; SIM, structured illumination microscopy; TA, tail-anchored; TEM, transmission electron microscopy; TIM, translocase of the inner mitochondrial membrane; TMD, transmembrane domain; TOM, translocase of the outer membrane; YFP, yellow fluorescent protein.

recognized and bound by the cytosolic targeting factor Pex5 and translocated via interaction of Pex5 with Pex13/14 on the peroxisomal membrane allowing the translocation of folded and even oligomerized protein complexes through Pex13, which acts as a pore resembling the nuclear import system [4–8]. Peroxisomes can multiply by growth and division, but unlike mitochondria can also form de novo in the absence of mature peroxisomes from the endoplasmic reticulum (ER) in a process which possibly involves mitochondria, at least in mammalian cells [4,9,10].

For many years, proteins were studied that either have a C-terminal PTS1 or an N-terminal mitochondrial targeting signal (MTS). Recently, it was shown that the protein phosphatase Ptc5 from the yeast *Saccharomyces cerevisiae* contains both targeting signals and exhibits a dual localization to peroxisomes and mitochondria [11]. Ptc5 is processed by the inner membrane peptidase (IMP) complex inside mitochondria and reaches the peroxisomal matrix via mitochondrial transit [11,12]. Additional proteins containing competing mitochondrial and peroxisomal targeting signals have been identified. Overexpression of several enhanced the number of peroxisomes proximal to mitochondria [11]. Data from mammals point to a similar phenomenon [13].

Organelle tethering has been a subject of intense research in recent years as it has become clear that organelles do not work in isolation but rather form areas of close apposition, contact sites, that enable the transfer of molecules and ions [14]. Tethering molecules that sustain such contacts have been discovered and, until now, have been shown to rely on components that either span the membrane or tightly bind to it [15]. Such a contact site is formed between peroxisomes and mitochondria and has been dubbed the PerMit [16,17]. Genetic data and protein interaction experiments reveal that the membrane proteins Pex11, Pex34, Fzo1, and the ERMES complex are important for peroxisome–mitochondria tethering [17,18]. Other tethers for peroxisomes to different organelles have been identified [19–25]. In mammalian cells, proximity of peroxisomes and the ER is achieved through interaction of membrane-associated proteins called VAMP-associated proteins A and B with a peroxisomal protein termed acyl-CoA binding domain-containing 5 [26,27]. Deletion of a single tether does not result in the loss of contact sites, suggesting that multiple tethering mechanisms are involved in the formation of each contact.

Here, we report on proteins with targeting signals for at least 2 organelles, which act as tethers through affinity for 2 targeting/translocation machineries.

## Results

### Ultrastructural analysis of contacts induced by overexpression of Ptc5-GFP-PTS in *Ustilago maydis*

Many proteins contain targeting signals for mitochondria and peroxisomes at opposite termini [11]. We hypothesized that this protein composition can contribute to the formation of organelle contact sites. We first addressed this concept by investigating the putative Ptc5 homolog of *Ustilago maydis* (Fig 1A) [11], a fungus that has many peroxisomes and is well-suited for ultrastructural analysis [28–30]. We initially analyzed strains overexpressing *U. maydis* Ptc5 fused to GFP but retaining the PTS1 (Um_Ptc5-GFP-PTS) via epifluorescence (Figs 1B and S1A) and structured illumination microscopy (SIM) (S1B Fig). Overexpression of Um_Ptc5-GFP-PTS significantly increased PerMit contacts (Fig 1B). A control protein Um_Ptc5-GFP, which lacked the PTS1, did not induce this phenotype. We went on to characterize contacts of mitochondria and peroxisomes by transmission electron microscopy (TEM). To visualize peroxisomes, we performed immunogold labeling to stain the peroxisomal marker mCherry-PTS. Enrichment of peroxisomes adjacent to mitochondria was observed

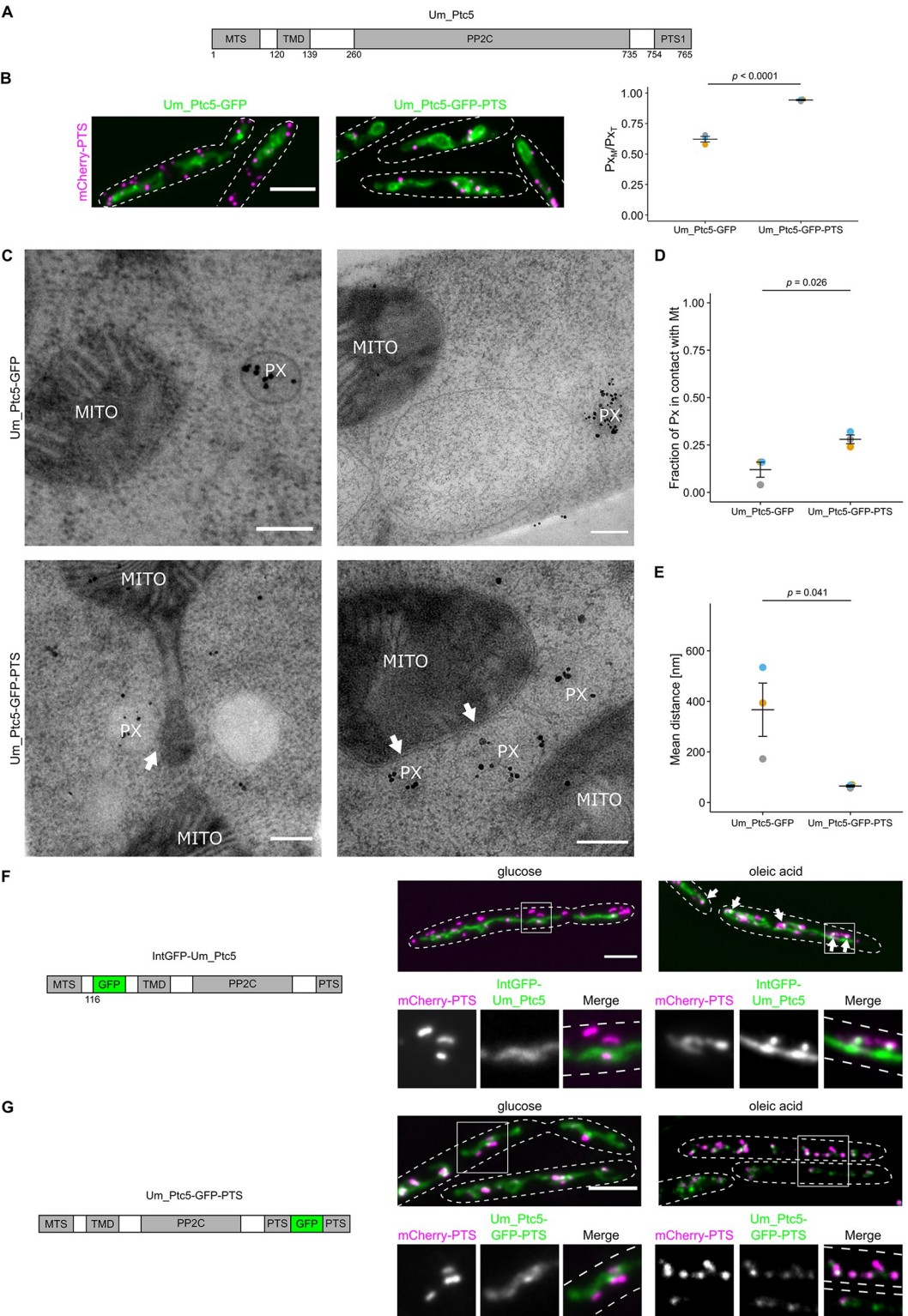

**Fig 1. Ultrastructural characterization of Um_Ptc5 induced contacts. (A)** Scheme of *U. maydis* Ptc5 highlighting the targeting signals (MTS, PTS), a TMD and the phosphatase domain (PP2C). Prediction of the PP2C domain was performed with the HMMER web server. Note that this sequence contains a putative cleavage site between TMD and PP2C, which has been experimentally validated only in yeast so far [11,12]. **(B)** Cells expressing Um_Ptc5-GFP or Um_Ptc5-GFP-PTS (green) under control of the constitutive *Otef*-promoter [33] and together with the peroxisomal marker mCherry-PTS (magenta)

were analyzed by fluorescence microscopy. **(C)** Pictures were obtained by TEM. mCherry-PTS was stained via immunogold labeling. White arrows indicate PerMit contacts. Scale bar: 0.2 μm. **(D)** and **(E)** Quantifications are based on $n = 3$ experiments. Each color represents 1 experiment. At least 25 peroxisomes per experiment were analyzed. Error bars represent SEM. *P*-values were calculated using a two-sided unpaired Student's *t* test. **(F)** Cells expressing an internally GFP-tagged derivative of Um_Ptc5 (green) under control of the endogenous promoter together with the peroxisomal marker mCherry-PTS (magenta) were analyzed by fluorescence microscopy following incubation with indicated carbon sources. Insets show single channels and merged channels. Scale bar: 5 μm. **(G)** Cells expressing a C-terminally GFP-tagged derivative of Um_Ptc5 preserving the PTS (green) under the control of the endogenous promotor and the peroxisomal marker mCherry-PTS (magenta) were analyzed by fluorescence microscopy following incubation with indicated carbon sources. Insets show single channels and merged channels. Scale bars in fluorescence microscopic images: 5 μm. Underlying data for quantifications can be found in S1 Data. MTS, mitochondrial targeting signal; PTS, peroxisome targeting signal; TEM, transmission electron microscopy; TMD, transmembrane domain.

upon overexpression of Um_Ptc5-GFP-PTS (Fig 1C and 1D). We quantified the mean distance of both organelles and detected a significant decrease in the strain containing Um_Ptc5-GFP-PTS compared to a control strain containing Um_Ptc5-GFP (Fig 1E). Hence, quantifications of proximity in EM data were consistent with quantifications in fluorescence microscopic pictures (Fig 1B) suggesting that the Ptc5 fusion protein acted as a tether.

Since those experiments involved overexpression, we performed an additional assay to confirm that Ptc5 accumulates at contact sites. Previously, we demonstrated that Ptc5 dephosphorylates the glycerol-3-phosphate dehydrogenase Gpd1 in yeast, a protein relevant for shuttling NADH [11,29,31,32].

We anticipated that growth on oleic acid might modulate transit and tethering since this condition is likely to require NADH shuttling. Cells containing an internally GFP tagged version of *Um_Ptc5* under control of the endogenous promoter were incubated in oleic acid medium (Fig 1F). Under these conditions, the GFP fusion protein accumulated in focal structures at the interface of peroxisomes and mitochondria (Figs 1F and S1C), which may represent sites of tethering; however, no translocation to peroxisomes was observed. This notion was reinforced by SIM experiments using identical conditions (S1D Fig). Transit may be affected by internal tagging. Thus, we also created an Um_Ptc5 variant containing a C-terminal GFP tag in which the C-terminal dodecamer was retained and expressed this construct under control of the endogenous promoter. Upon incubation in oleic acid medium, we observed enhanced colocalization of Ptc5 with peroxisomes (Figs 1G and S1E), suggesting that targeting of Ptc5 is regulated in a condition that requires peroxisome function.

## Proteins that contain both an MTS and a PTS1 increase proximity of mitochondria and peroxisomes upon overexpression in *S. cerevisiae*

To gain more insight into sorting and tethering via dual affinity proteins, we turned our attention to *S. cerevisiae* (from hereon called yeast), which expresses a diverse group of proteins containing targeting signals for peroxisomes and mitochondria [11,34] (Fig 2A and S1 Table). We tested tethering by overexpressing several candidate proteins fused to a red fluorescent protein (RFP) at their C-terminus, in a manner preserving the original PTS1. We consistently detected an elevated number of peroxisomes in proximity to mitochondria (Figs 2B–2D, S2A, and S2B) [11]. In addition, overexpression of 2 candidates—Cat2 (carnitine acetyltransferase)-RFP-HA-PTS and Pxp2-RFP-HA-PTS—reduced the number of peroxisomal foci compared to control strains lacking the PTS1, a phenotype previously observed for a synthetic tether (Fig 2C and 2E) [17].

We were concerned that dual targeting and the enhanced proximity between the organelles may have depended on the use of a bulky tag such as RFP which can alter sorting and tethering (Fig 1F and 1G) [11]. To test this, we internally Myc-tagged Cat2, Cta1, Pxp2, and Tes1 at their

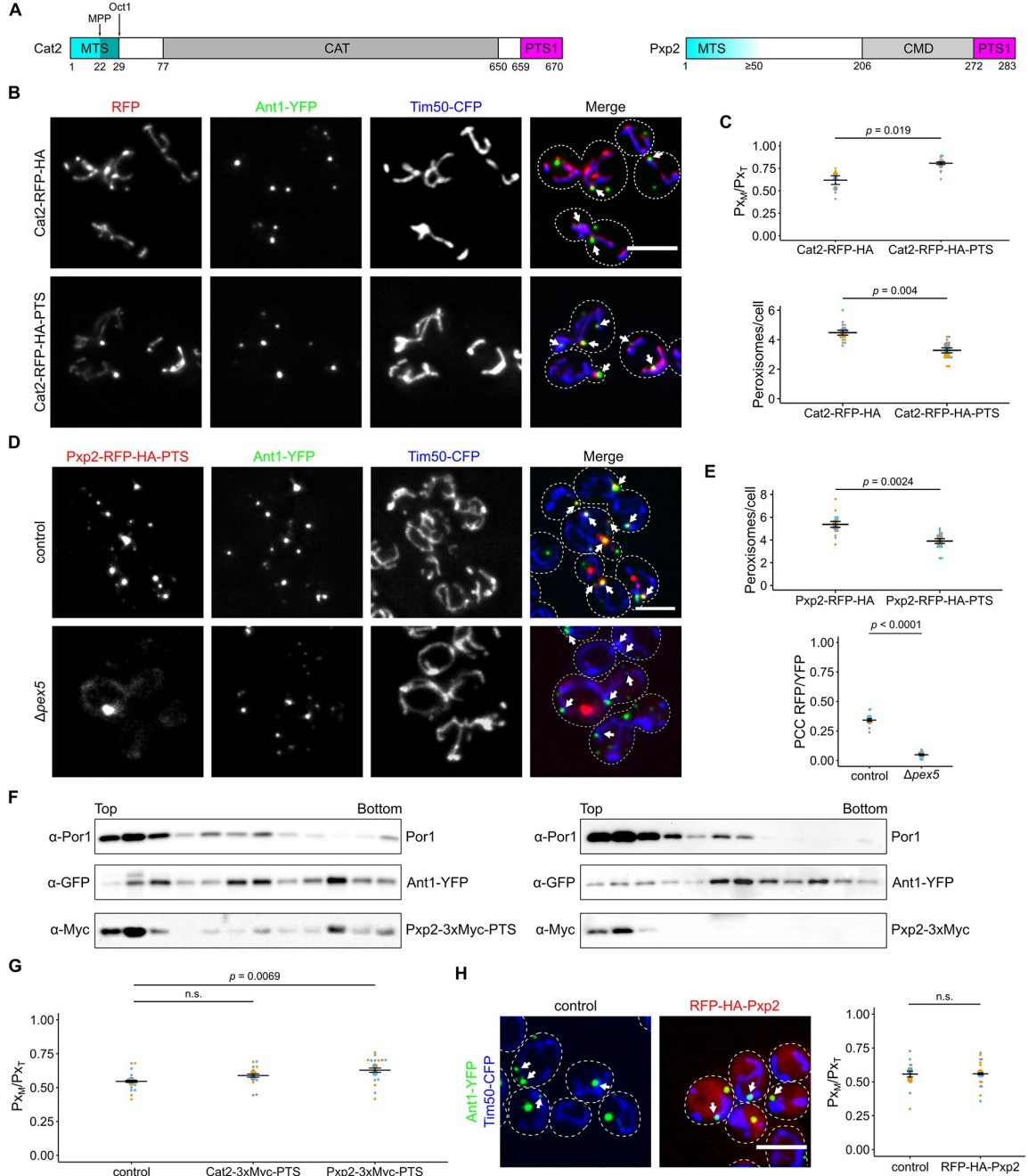

**Fig 2. Cat2 and Pxp2 localize to mitochondria and peroxisomes and induce interorganellar contacts. (A)** Scheme showing Cat2 and Pxp2. Cat2 contains an N-terminal MTS that is likely processed sequentially by 2 different mitochondrial proteases (MPP, Oct1) [109] and contains an acetyl-carnitine transferase (CAT) domain and a PTS1 [110]. Pxp2 contains an N-terminal MTS, a putative carboxymuconolactone decarboxylase domain (CMD) and a PTS1 [11]. Prediction of protein domains was performed with the HMMER web server. **(B)** Fluorescence microscopic images of yeast cells expressing either Cat2-RFP-HA or Cat2-RFP-HA-PTS (red) together with the peroxisomal membrane protein Ant1-YFP (green) and the mitochondrial inner membrane protein Tim50-CFP (blue). White arrows denote peroxisomes overlapping mitochondria. **(C)** Quantification of the fraction of peroxisomes in contact with mitochondria ($Px_M$) relative to the total peroxisome number ($Px_T$) (upper plot) and quantification of the number of peroxisomes per cell (lower plot) of cells shown in (B). **(D)** Fluorescence microscopic pictures of control and $\Delta pex5$ cells co-expressing Pxp2-RFP-HA-PTS (red) together with Ant1-YFP (green) and Tim50-CFP (blue) (left). White arrows denote peroxisomes overlapping mitochondria. **(E)** Quantification of the number of peroxisomes per cell (upper plot) and the correlation between the Pxp2-RFP-HA-PTS signal and Ant1-YFP signal (lower plot) of cells shown in (D). PCC refers to Pearson's correlation coefficient. **(F)** Subcellular localization of endogenously tagged Pxp2-3xMyc-PTS and Pxp2-3xMyc was determined using density gradient centrifugation. Twelve fractions, collected from the top of the gradient, were analyzed by SDS–PAGE and immunoblot. Ant1-YFP is a

peroxisomal membrane protein and Por1 is localized in the mitochondrial outer membrane. **(G)** Quantification of the fraction of peroxisomes contacting mitochondria ($Px_M$) in relation to the total peroxisome count ($Px_T$) of control cells and cells overexpressing the indicated fusion proteins. **(H)** Yeast cells expressing N-terminally tagged Pxp2 (red) together with Ant1-YFP (green) and Tim50-CFP (blue) were analyzed by fluorescence microscopy. The progenitor strain without RFP-HA-Pxp2 served as control. White arrows denote peroxisomes overlapping mitochondria. The fraction of peroxisomes in contact with mitochondria ($Px_M$) relative to the total peroxisome number of peroxisomes was plotted (right). Single-channel pictures are shown in S2D Fig. Quantifications are based on $n$ = 3 experiments. Each color represents 1 experiment. Error bars represent SEM. *P*-values were calculated using a two-sided unpaired Student's *t* test. For plots showing multiple comparison, a one-way ANOVA combined with a Tukey test was performed. Scale bars represent 5 μm. Underlying data for quantifications can be found in S1 Data. MTS, mitochondrial targeting signal; PTS, peroxisome targeting signal.

endogenous loci maintaining their original N- and C-termini to assess dual localization under more physiological conditions. Dual targeting was confirmed by evaluation of compartments resolved on buoyant density gradients (Figs 2F and S2C). We then tested overexpressed internally Myc-tagged versions of 2 candidate proteins: Cat2 and Pxp2. Even without the RFP tag, Pxp2-3xMyc-PTS and Cat2-3xMyc-PTS both increased the association of peroxisomes and mitochondria (Fig 2G) with Pxp2 showing a stronger effect.

A difference between both proteins was also reflected by their subcellular distribution. Whereas Cat2-RFP-HA-PTS was found to localize distributed over mitochondria and in peroxisomes, Pxp2-RFP-HA-PTS occurred in foci at mitochondria frequently proximal to a peroxisome but also colocalized with peroxisomes without contact to mitochondria (Fig 2B and 2D). Pxp2-RFP-HA-PTS became more evenly distributed in mitochondria upon deletion of *PEX5* (Fig 2D). To assess if enhanced tethering depended on the presence of the MTS, we overexpressed an N-terminal RFP-HA tagged variant of Pxp2 to mask the MTS. This fusion protein colocalized with a peroxisomal marker but also occurred in the cytosol and did not induce PerMit contacts (Figs 2H and S2D). Thus, overexpression of proteins containing both a mitochondrial and a peroxisomal targeting signal increases association of both organelles in yeast.

## Loss of the matrix protein targeting machinery reduces the number of contacts between peroxisomes and mitochondria

If tethering is affected by dual targeting of MTS-PTS proteins, it should be reduced by eliminating the targeting factor Pex5. To test this, we quantified the number of peroxisomes associated with mitochondria in control and Δ*pex5* cells. A significant reduction of peroxisomes close to mitochondria was observed in the absence of Pex5 (Figs 3A, 3B, and S3) indicating a role of the peroxisomal matrix targeting machinery for the generation of contacts—something unexpected if only membrane proteins were required for tethering as is usually assumed. To further confirm this concept, we deleted *PEX5* in a strain overexpressing Pxp2-RFP-PTS. Enhanced tethering triggered by Pxp2-RFP-PTS was blocked by deleting *PEX5* (Fig 3C). In addition, we quantified the peroxisome number and detected an increase of smaller appearing peroxisomes upon deletion of *PEX5* indicating that the reduced colocalization is not a result of a reduction in peroxisome number (Figs 3D and S3C). Another phenotype previously described for mutants with reduced peroxisome tethering is increased movement of the organelles [19,26]. Peroxisome movement was found to be enhanced in Δ*pex5* cells (S1 Movie). These results indicate that overexpression of dual affinity MTS-PTS proteins and depletion of Pex5 have a reciprocal effect on peroxisome number, morphology, and proximity to mitochondria.

To confirm our findings and reduce the possibility that the observed phenotype only resulted from secondary, downstream effects, we took an approach for rapid depletion of the Pex5 receptor, Pex13 [6], using an auxin-inducible degron (AID) tag [35,36] (Fig 3E). Peroxisomal import of RFP-PTS was still visible upon AID-HA tagging of Pex13 demonstrating that the tagging itself does not abolish import. Addition of auxin lead to cytosolic accumulation of

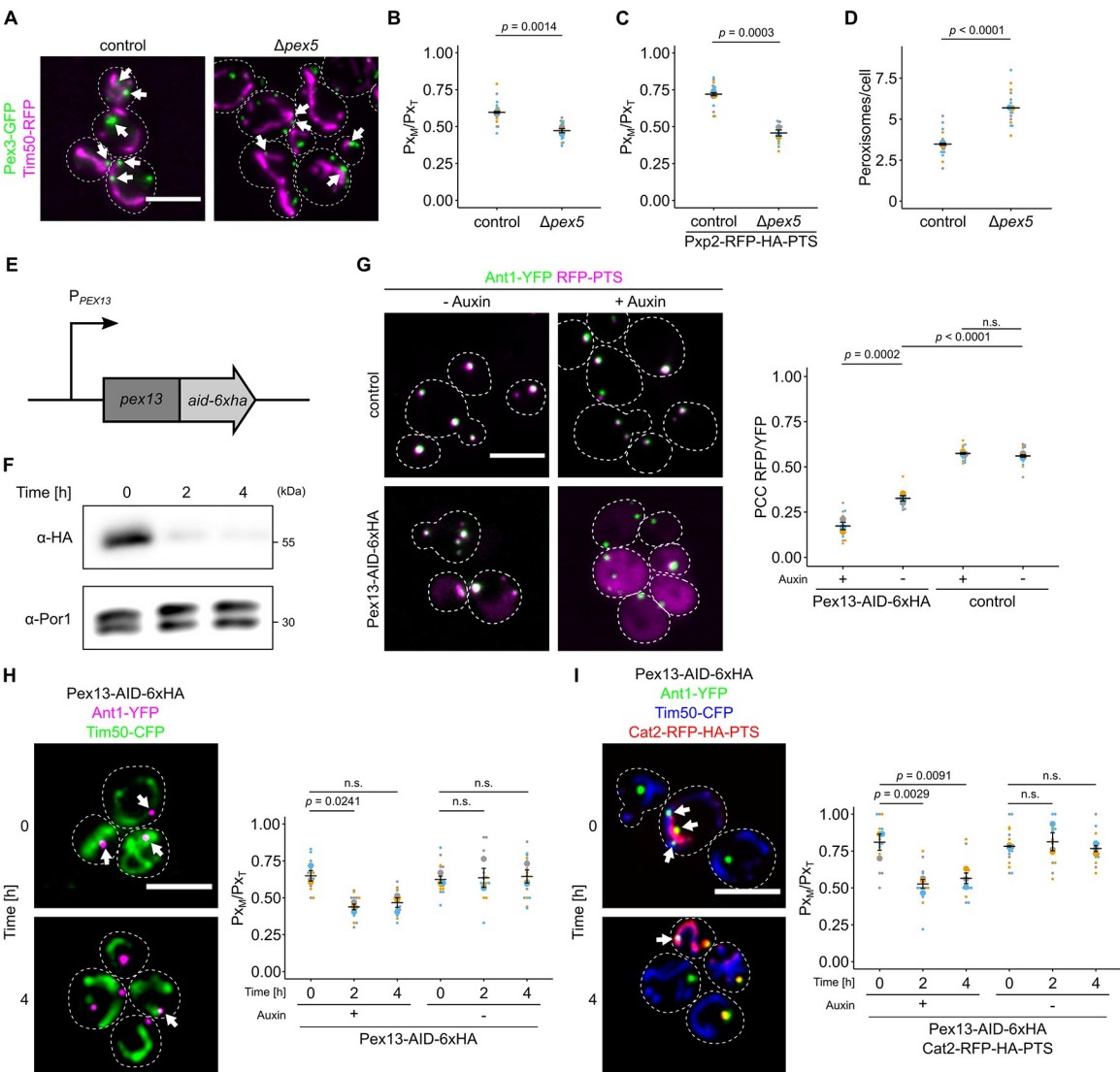

**Fig 3. Depletion of components of the peroxisomal import machinery reduces PerMit contacts. (A)** Fluorescence microscopic picture of control and Δ*pex5* cells expressing endogenously tagged Pex3-GFP (green) and Tim50-RFP (magenta). White arrows denote peroxisomal signal overlapping with mitochondrial signal. **(B)** Quantification of the fraction of peroxisomes contacting mitochondria ($Px_M$) in relation to the total peroxisome count ($Px_T$) of control cells and Δ*pex5* cells. **(C)** Quantification of the fraction of peroxisomes contacting mitochondria ($Px_M$) in relation to the total peroxisome count ($Px_T$) of control cells and Δ*pex5* cells expressing Pxp2-RFP-PTS, Ant1-YFP, and Tim50-CFP. **(D)** The number of peroxisomes per cell was quantified in the indicated strains expressing Pex3-GFP. **(E)** Scheme of the genetic modifications used for auxin-dependent depletion of Pex13 in (F)–(I). The endogenous *PEX13* locus was genetically engineered to encode a translational fusion of Pex13 with a C-terminal AID and 6 hemagglutinin (HA) tags. Pex13 degradation is mediated by the F-box protein AFB2 from *Arabidopsis thaliana*, which was expressed from the *ADH1* promotor. **(F)** Auxin-dependent depletion of Pex13-AID-HA at indicated time points was analyzed by SDS-PAGE and immunoblot. Por1 served as a loading control. **(G)** Fluorescence microscopic images of indicated strains expressing the peroxisomal membrane protein Ant1-YFP (green) and RFP-PTS (magenta) in the absence (-Auxin) or presence (+Auxin; 4 h) of 2 mM indole-3-acetic acid. **(H)** Subcellular localization of Ant1-YFP (magenta) and Tim50-CFP (green) of indicated strains was analyzed in the presence of 2 mM indole-3-acetic acid at indicated time points (left). White arrows indicate peroxisomes in proximity to mitochondria. The fraction of peroxisomes in contact with mitochondria ($Px_M$) relative to the total peroxisome count ($Px_T$) of the indicated strain was quantified (right). **(I)** Identical to (H), except that the cells also expressed Cat2-RFP-HA-PTS (red) to increase PerMit contacts. Scale bars represent 5 μm. Quantifications are based on $n = 3$ experiments. Each color represents 1 experiment. Error bars represent SEM. A one-way ANOVA combined with a Tukey test was performed to assess statistical significance for multiple comparisons. Otherwise, a two-sided unpaired Student's *t* test was performed. Underlying data for quantifications can be found in S1 Data. AID, auxin-inducible degron; PTS, peroxisome targeting signal; RFP, red fluorescent protein.

RFP-PTS (Figs 3G and S4A) and reduction of Pex13-AID-HA levels (Fig 3F). Association of peroxisomes and mitochondria was quantified in strains with or without Cat2-RFP-HA-PTS as an additional tether relative to control strains (Figs 3H, 3I, S4B–S4D, and S5). Both strains started with a different degree of organelle association but depletion of Pex13-AID-HA by auxin addition reduced the number of peroxisomes proximal to mitochondria to a similar basal level. These differences in tethering support a role of dual protein targeting in driving organelle association.

To corroborate these data, we created a strain with conditional *PEX5* expression by putting it under control of the *GAL* promoter (S6 Fig). Immunoblot and fluorescence microscopy of RFP-PTS expressing cells confirmed that Pex5 levels correlated with PTS import (S6A and S6B Fig). We then overexpressed Ptc5ΔTMD-RFP-PTS, a dually targeted derivative of Ptc5 lacking the transmembrane domain (TMD) that acts as a strong tether [11]. Automated time lapse imaging revealed that induction of Pex5 elevated the number of peroxisomes close to mitochondria, whereas its reduction had the opposite effect (S6C Fig). Since single planes were imaged, lower basal overlap of mitochondria and peroxisomes was observed in the absence of Pex5 (compare to Fig 3B).

Together, these data suggest a function of peroxisomal matrix protein import for formation of PerMit contacts.

## The dual affinity protein Lys12 regulates PerMit contacts upon lysine deprivation

Peroxisomes participate in lysine biosynthesis in several fungi [37–39]. The last step catalyzed by saccharopine dehydrogenase is located in the matrix of yeast peroxisomes [32,37]. Of interest, 2 other enzymes required for this pathway contain both an MTS and a putative PTS1 (Fig 4A and S1 Table). We verified that the PTS1 motifs of Lys4 (homoaconitase) and Lys12 (homo-isocitrate dehydrogenase) were functional by checking the subcellular localization of RFP fused to the C-terminal dodecamer of either protein (Fig 4B). Next, we tested an overexpressed version of Lys4 and Lys12 that either contained or lacked the PTS1. Overexpression of Lys12-RFP-HA-PTS significantly elevated the number of peroxisomes close to mitochondria; however, no translocation to peroxisomes was observed (Fig 4C). Increased expression of Lys4-RFP-HA-PTS was less effective but showed a similar trend (Fig 4D).

It was known that lysine deprivation enhances expression of lysine biosynthesis proteins [37]. To confirm these data, we created endogenously Myc-tagged versions of Lys12. Expression of Lys12 was enhanced upon lysine deprivation (Fig 4E). To test whether this alters proximity of peroxisomes and mitochondria, we compared the number of peroxisomes close to mitochondria in cells incubated in medium with and without lysine. The number of contacts increased in cells lacking a supply of lysine (Fig 4F). Masking of the PTS1 of Lys12 by a C-terminal 9xMyc-tag at the endogenous locus abolished this increased tethering (Fig 4F). Reintroduction of the PTS1 at the genomic locus rescued the mutant phenotype (Fig 4G). We did not observe a growth defect for cells lacking a functional PTS1 in Lys12 (S7 Fig); however, this was not surprising as even cytosolic mistargeting of Lys1, e.g., via deletion of *PEX5* does not drastically interfere with lysine biosynthesis (S7 Fig) [32]. In aggregate, these data show that tethering via dual affinity proteins is a regulated process and depends on the metabolic state of the cell. In addition, these results indicate that translocation to peroxisomes is no prerequisite for tethering via dual affinity proteins. Retention in mitochondria may even promote tethering.

## Screening for factors that regulate dual targeting of Ptc5

To identify additional factors that regulate dual targeting and therefore potentially also tethering, we conducted a high-content genetic screen focused on Ptc5 sorting from mitochondria

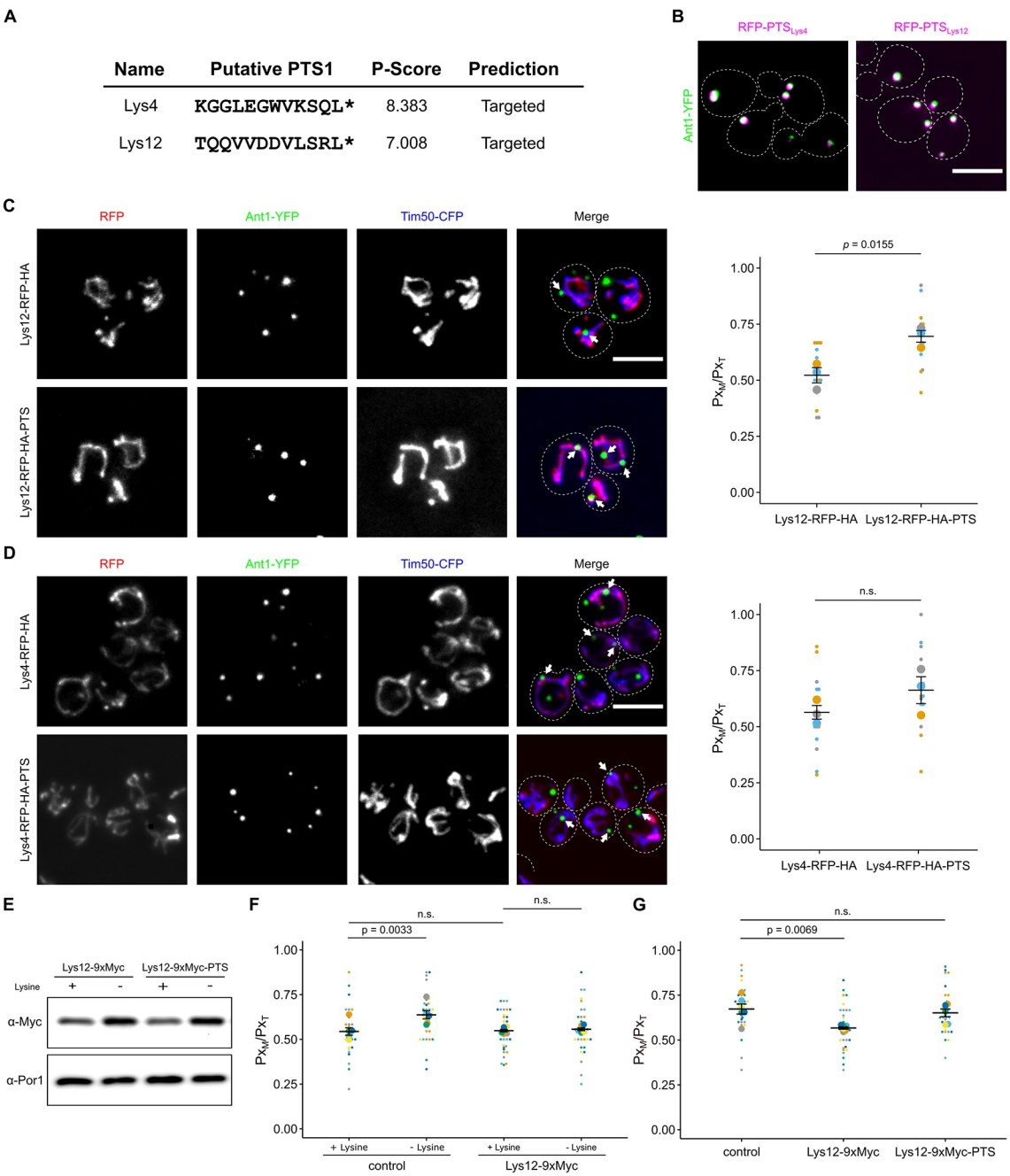

**Fig 4. The dual affinity protein Lys12 enhances PerMit contacts upon lysine deprivation. (A)** PTS1 motifs in mitochondrial enzymes Lys4 and Lys12 were predicted using a program from Neuberger and colleagues [40]. **(B)** C-terminal dodecamers depicted in Fig 4A were fused to RFP (magenta) and tested for peroxisomal localization in strains expressing Ant1-YFP (green). **(C)** Fluorescence microscopic images of yeast cells expressing either Lys12-RFP-HA or Lys12-RFP-HA-PTS (red) together with the peroxisomal membrane protein Ant1-YFP (green) and the mitochondrial inner membrane protein Tim50-CFP (blue) (left). White arrows denote peroxisomes overlapping mitochondria. Quantification of the fraction of peroxisomes in contact with mitochondria ($Px_M$) relative to the total peroxisome number ($Px_T$) (right). **(D)** Identical to Fig 4C for Lys4. **(E)** Immunoblot showing the levels of indicated fusion protein from cells grown with or without lysine. **(F)** Quantification of the fraction of peroxisomes contacting mitochondria ($Px_M$) in relation to the total peroxisome count ($Px_T$) of control cells and cells endogenously expressing a C-terminally 9xMyc-tagged derivative, which masks the PTS1 of Lys12 grown with or without lysine. **(G)** Quantification of the fraction of peroxisomes contacting mitochondria ($Px_M$) in relation to the total peroxisome count ($Px_T$) of control cells and cells endogenously expressing a C-terminally 9xMyc-tagged derivative, which either masks or preserves the PTS1 of Lys12 grown with or without lysine. Quantifications are based on $n$ = 3 experiments. Each color represents 1 experiment. Error bars represent SEM. *P*-values were calculated using a two-sided unpaired Student's *t* test. For plots showing multiple comparison, a one-way ANOVA combined with a Tukey test was performed.

Scale bars represent 5 μm. Underlying data for quantifications can be found in S1 Data. PTS, peroxisome targeting signal; RFP, red fluorescent protein.

to peroxisomes (Fig 5A). Peroxisomal import of Ptc5-RFP-PTS depends on mitochondrial processing and subsequent Pex5 dependent targeting [11]. This transit mechanism may require direct contact between the organelles.

Crossings of a control strain expressing Ptc5-RFP-PTS and Pex3-GFP as a peroxisomal marker with arrayed libraries containing yeast deletion strains and hypomorphic mutants of essential genes were performed and sporulated to select for haploid cells containing both fluorescent proteins together with a genetic perturbation (Fig 5A) [41–43]. Inspection of haploid progeny by automated microscopy uncovered various genes affecting peroxisomal targeting of Ptc5-RFP-PTS (Figs 5B–5D and S8 and S2 Table). These include many expected genes involved in peroxisome biogenesis but also distinct genes, e.g., *SOM1* and *IMP1*. Som1 was shown previously to be part of the IMP complex [44], which we could confirm (Figs 5C and S8).

## Depletion of ERMES affects the distribution of dually targeted proteins

Looking at the screen results, we found that an interesting gene whose deletion affected the co-localization between Ptc5 and peroxisomes was *MDM10*. Mdm10 is a critical factor for forming a contact site between mitochondria and the ER as part of the ER–mitochondria encounter structure (ERMES) complex (Fig 6A) [45]. The ERMES complex consists of the ER-anchored protein Mmm1 and 3 mitochondrial proteins (Mdm10, Mdm12, and Mdm34) [45]. The integral membrane protein Mdm10 has an additional function and controls import of mitochondrial ß-barrel proteins [46]. It was therefore unclear whether the effect of Mdm10 on Ptc5 localization was through the ERMES or by affecting mitochondrial protein import.

To address whether the observed phenotype is specific to Mdm10 or shared by other ERMES complex members, we repeated our assay in freshly made strains that have reduced levels of possible suppressor mutations. In all ERMES mutants targeting of Ptc5-RFP-PTS to peroxisomes was significantly reduced (Figs 6B and S9) with Δ*mdm10* displaying the strongest phenotype and Δ*mmm1* being least affected.

To test the stage at which ERMES mutants affected dually targeted proteins, we first analyzed the initial steps of mitochondrial protein import. All mutants influenced mitochondrial import and preprotein processing as determined by testing a truncated variant of Ptc5-RFP (Fig 6C). Moreover, we observed an import defect for several bona fide mitochondrial proteins in Δ*mdm10* cells (S10A Fig) demonstrating that loss of ERMES complex altered translocation of mitochondrial proteins in a more general way. If compromised initial protein targeting of Ptc5-RFP-PTS to mitochondria was responsible for reduced peroxisomal targeting, mutants lacking components of the mitochondrial import machinery should exhibit a similar phenotype. As these were not identified in our screen, we tested if these were false negatives. We assessed Ptc5-RFP-PTS localization in Δ*tom22*, Δ*tom70*, and in a mutant lacking the mitofusin, Fzo1 that contains defective mitochondria similar to Δ*mdm10* (S10B Fig). None of these mutants showed mitochondrial retention of Ptc5-RFP-PTS, but preprotein processing defects were observed in *FZO1* depleted cells (S10B and S10C Fig). In Δ*tom70* cells, a reduced but peroxisomal signal of Ptc5 was observed (S10B and S10C Fig), demonstrating that perturbation of the initial steps in mitochondrial import is not sufficient to block Ptc5 targeting to peroxisomes. The previously suggested binding partner of ERMES Pex11 [18] did not interfere with Ptc5 transit. The phenotype of ERMES mutants is therefore distinctive.

To follow up on these results, we determined the localization of other proteins that contain competing targeting signals in Δ*mdm10* cells. Cat2-RFP-PTS behaved similarly to

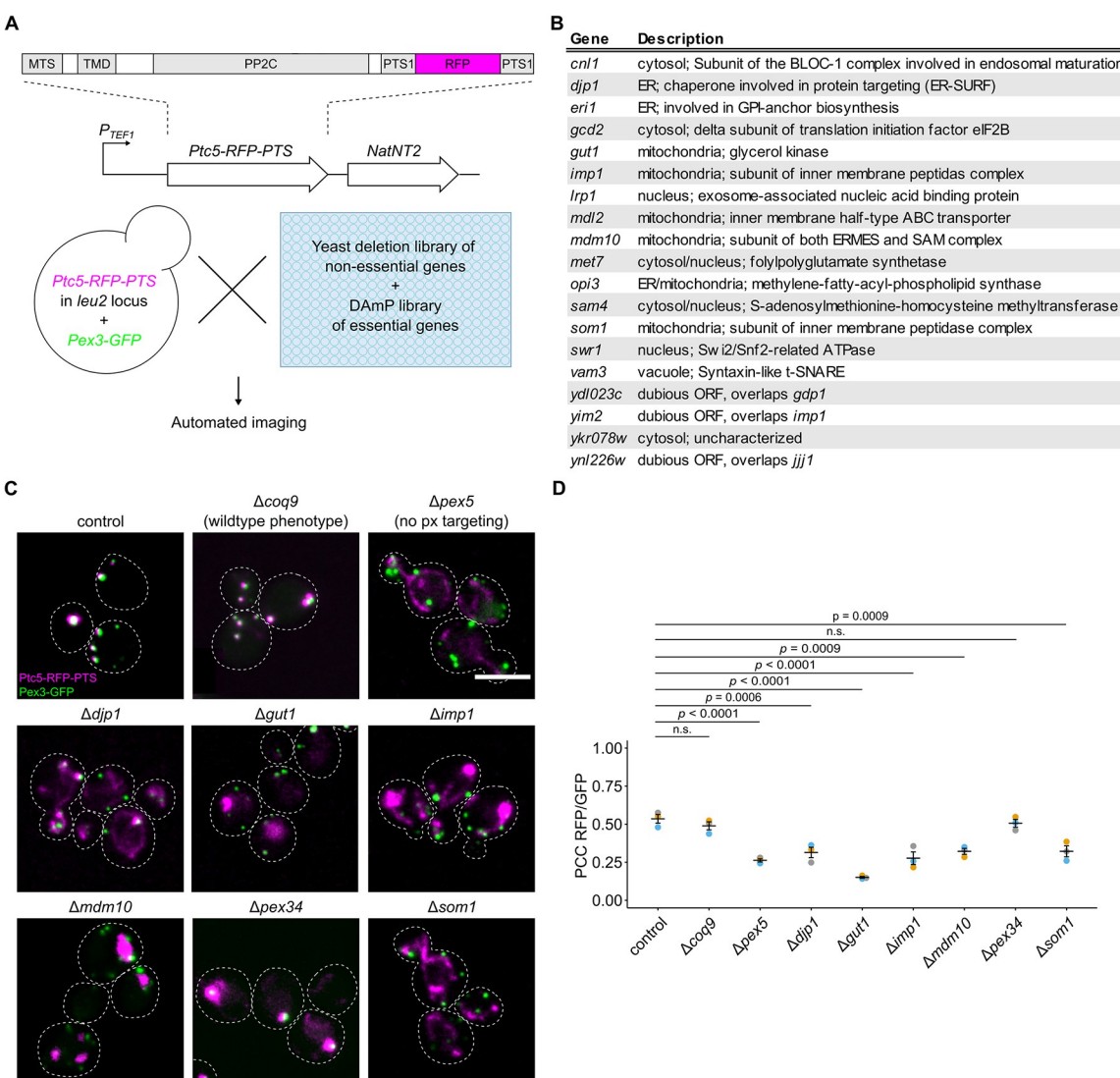

**Fig 5. A high-content genetic screen to identify factors involved in targeting of Ptc5 to peroxisomes. (A)** Schematic illustration of a high-content microscopic screen to uncover factors involved in sorting of the protein phosphatase Ptc5. A query strain co-expressing Ptc5-RFP-PTS and Pex3-GFP was crossed into indicated yeast libraries using synthetic genetic array technology. Haploid progeny expressing both fusion proteins in each mutant background were analyzed with automated fluorescence microscopy. MTS, mitochondrial targeting signal; TMD, transmembrane domain; PP2C, protein phosphatase type 2C; PTS1, peroxisomal targeting signal type 1. **(B)** Table showing a subset of mutants affecting Ptc5 localization. The entire list of mutations is depicted in S2 Table. **(C)** Subcellular localization of Ptc5-RFP-PTS (magenta) and Pex3-GFP (green) was analyzed in indicated strains using automated fluorescence microscopy. Shown are pictures from experiments performed on putative hits. Δ*coq9* cells were used as a control as these show Ptc5-RFP-PTS localization in peroxisomes but are affected in mitochondrial metabolism [11]. Δ*pex5* mutants show mitochondrial but no peroxisomal signal of the reporter Ptc5-RFP-PTS. Scale bars represent 5 μm. **(D)** Correlation between Ptc5-RFP-PTS signal and Pex3-GFP signal was quantified in indicated strains. Quantifications are based on *n* = 3 experiments. Each color represents 1 experiment. Error bars represent SEM. A one-way ANOVA combined with a Tukey test was performed to assess statistical significance. Underlying data for quantifications can be found in S1 Data. PTS, peroxisome targeting signal; RFP, red fluorescent protein.

Ptc5-RFP-PTS and was retained in mitochondria while targeting to peroxisomes was reduced (Figs 6D and S11A). In contrast, the localization of Pxp2-RFP-PTS, Tes1-RFP-PTS, and Mss2-RFP-PTS displayed the opposite pattern of localization in Mdm10 depleted cells with reduced mitochondrial targeting but normal peroxisome targeting (Figs 6E and S11B). Proteolytic processing defects could not be observed for Pxp2, Tes1, and Mss2 suggesting a

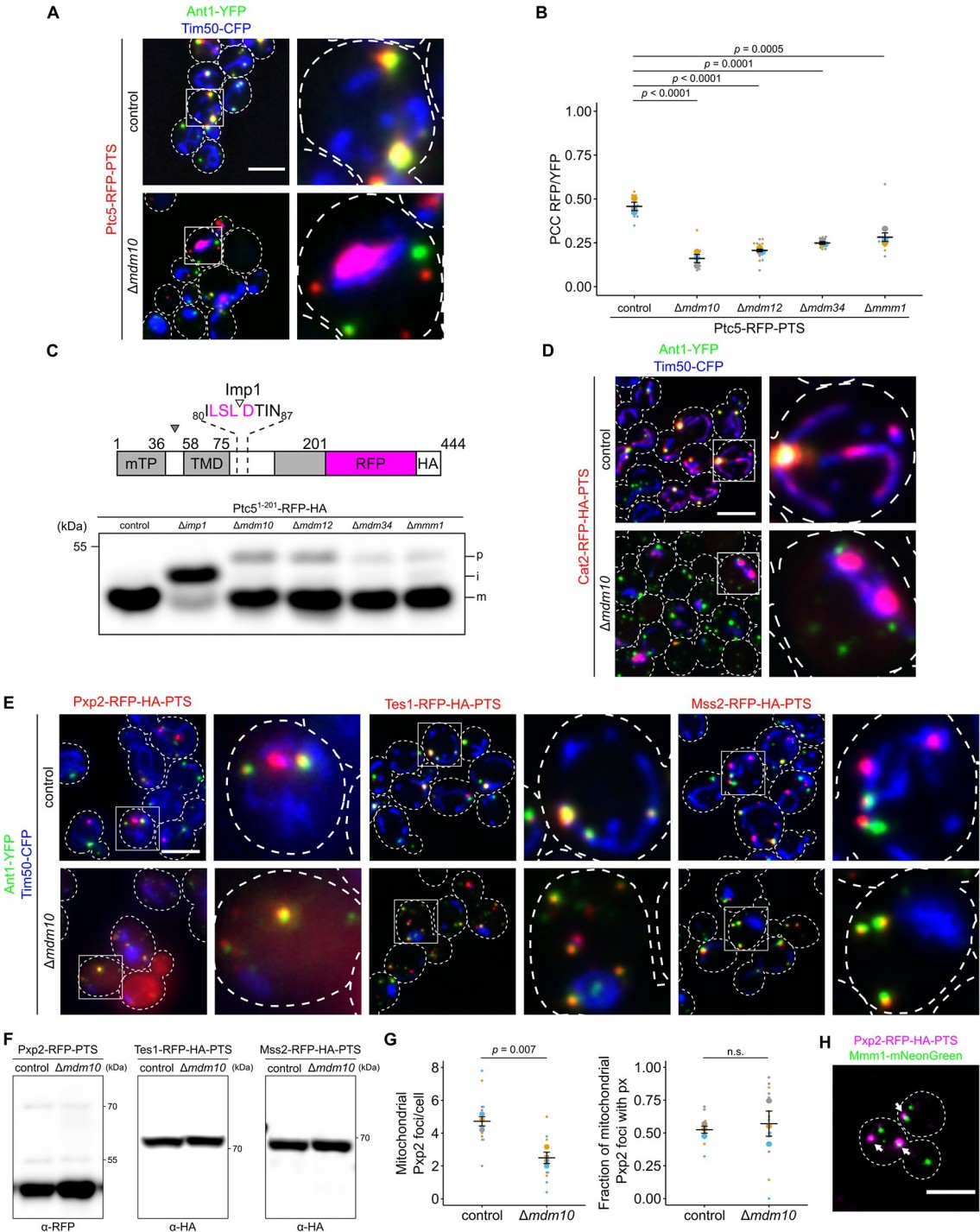

**Fig 6. ERMES complex regulates import of proteins into mitochondria and peroxisomes. (A)** Subcellular localization of Ptc5-RFP-PTS (red), the peroxisomal membrane protein Ant1-YFP (green), and the mitochondrial inner membrane protein Tim50-CFP (blue) in control or Δ*mdm10* cells was analyzed with fluorescence microscopy. **(B)** Correlation between Ptc5-RFP-PTS signal and Ant1-YFP signal was quantified in indicated strains. **(C)** The truncated variant Ptc5$^{1-201}$-RFP lacking a PTS1 was expressed in indicated strains. Cleavage sites for MPP (filled arrow) and the IMP complex (blank arrow) are indicated in the scheme. Whole cell lysates were analyzed by SDS-PAGE and immunoblot. p: premature isoform, i: intermediate isoform, m: mature isoform. Concentrations of protein extracts were adapted to each other to focus on processing. **(D)** Cat2-RFP-HA-PTS (red) was co-expressed with Ant1-YFP (green) and Tim50-CFP (blue) in control or Δ*mdm10* cells. Subcellular localization was determined with fluorescence microscopy. White arrows denote peroxisomes overlapping mitochondria. **(E)** Fluorescence microscopic pictures of indicated strains expressing respective RFP fusion proteins (red) together with Ant1-YFP (green) and Tim50-CFP (blue). **(F)** Whole cell lysates of

strains expressing the indicated fusion proteins were analyzed by SDS-PAGE and immunoblot. Concentrations of protein extracts were adapted to each other to focus on processing. **(G)** The number of Pxp2-positive foci per cell at mitochondria was quantified in indicated strains (left). Quantification of the fraction of mitochondrial Pxp2 foci overlapping Ant1-YFP (right). **(H)** Fluorescence microscopic picture of a strain expressing Pxp2-RFP-HA-PTS (magenta) together with Mmm1-mNeonGreen (green). White arrows indicate Pxp2-RFP-PTS foci overlapping Mmm1-mNeonGreen foci. Scale bars represent 5 μm. Quantifications are based on $n = 3$ experiments. Each color represents 1 experiment. Error bars represent SEM. *P*-values were calculated using a two-sided unpaired Student's *t* test. For multiple comparisons, *P*-values were calculated with a one-way ANOVA combined with a Tukey test. Underlying data for quantifications can be found in S1 Data. ERMES, endoplasmic reticulum–mitochondria encounter structure; IMP, inner membrane peptidase; PTS, peroxisome targeting signal; RFP, red fluorescent protein.

noncanonical MTS (Fig 6F). Quantification showed a reduction of mitochondrial foci containing Pxp2-RFP-PTS (Fig 6G). ERMES components could be directly involved in mitochondrial targeting as we found that Pxp2-RFP-PTS foci regularly (38% +/− 5%) overlapped with Mmm1-mNeonGreen foci (Fig 6H and S3 Table), a phenotype previously observed for a Per-Mit reporter, which also enhances contact when overexpressed [17]. Hence, contact forming proteins show a tendency to accumulate at ERMES sites. In aggregate, these data revealed ERMES as an important factor to determine the distribution of various dually targeted proteins. Interestingly, for different cargo either predominant peroxisomal targeting or mitochondrial targeting was observed.

## Depletion of ERMES affects proximity of peroxisomes and mitochondria

Since mitochondrial retention of Ptc5-RFP-PTS could not to be explained by mitochondrial dysfunction or defective mitochondrial protein import in ERMES mutants, we tested other possible reasons for this phenotype. To assess if it results from reduced ER–mitochondria connections, we tested if a synthetic ER–mitochondria tether [45] rescues the peroxisome-specific phenotypes of Δ*mdm10* cells. This was not the case (S12A Fig). Previously, it was shown that ERMES puncta frequently colocalize with peroxisomes and that overexpression of ERMES components induces PerMit contacts [16–18], a possible prerequisite for Ptc5 translocation. Alternatively, compromised import of PTS1 proteins at peroxisomes could explain Ptc5 retention in mitochondria. Peroxisomal targeting of the reporter RFP-PTS was only slightly affected (Figs 7A and S12B). Pxp2, Mss2, and Tes1 showed normal or even enhanced peroxisomal localization upon *MDM10* deletion (Fig 6E). In addition, a variant of Ptc5 lacking the MTS colocalized with peroxisomes in Δ*mdm10* cells (S12C Fig) indicating that it is not their inability to import PTS1 cargo, which causes mitochondrial accumulation of Ptc5-RFP-PTS.

As already observed by others [16,47], we found that ERMES mutants contained a higher number of small peroxisomes in a manner similar to a Δ*pex5* strain (Fig 7A and 7B). Furthermore, the fraction of peroxisomes in proximity to mitochondria declined in the ERMES mutants (Fig 7C). These phenotypes often reverted and, for example, we obtained Δ*mdm12* cells with wild-type–like growth (S13A Fig) and regular mitochondrial morphology (S13B Fig). Phenotypic suppression of ERMES mutants by second site mutations is regularly observed [48]. These suppressor strains still contained a higher number of small peroxisomes (S13C Fig).

Together, our data point to specific functions of ERMES for proper peroxisome formation as previously suggested [18,47]. ERMES may affect the proximity of mitochondria and peroxisomes—directly [18] or indirectly. Since we had noticed that overexpression of MTS-PTS proteins resulted in increased proximity and a reduced number of peroxisomes, we tested whether a synthetic tether, designed to connect peroxisomes and mitochondria (Fig 7D), affected peroxisomes in a manner that rescued loss of ERMES components. We found that in control but also Δ*mdm10* and Δ*mdm34* cells expression of a synthetic tether decreased

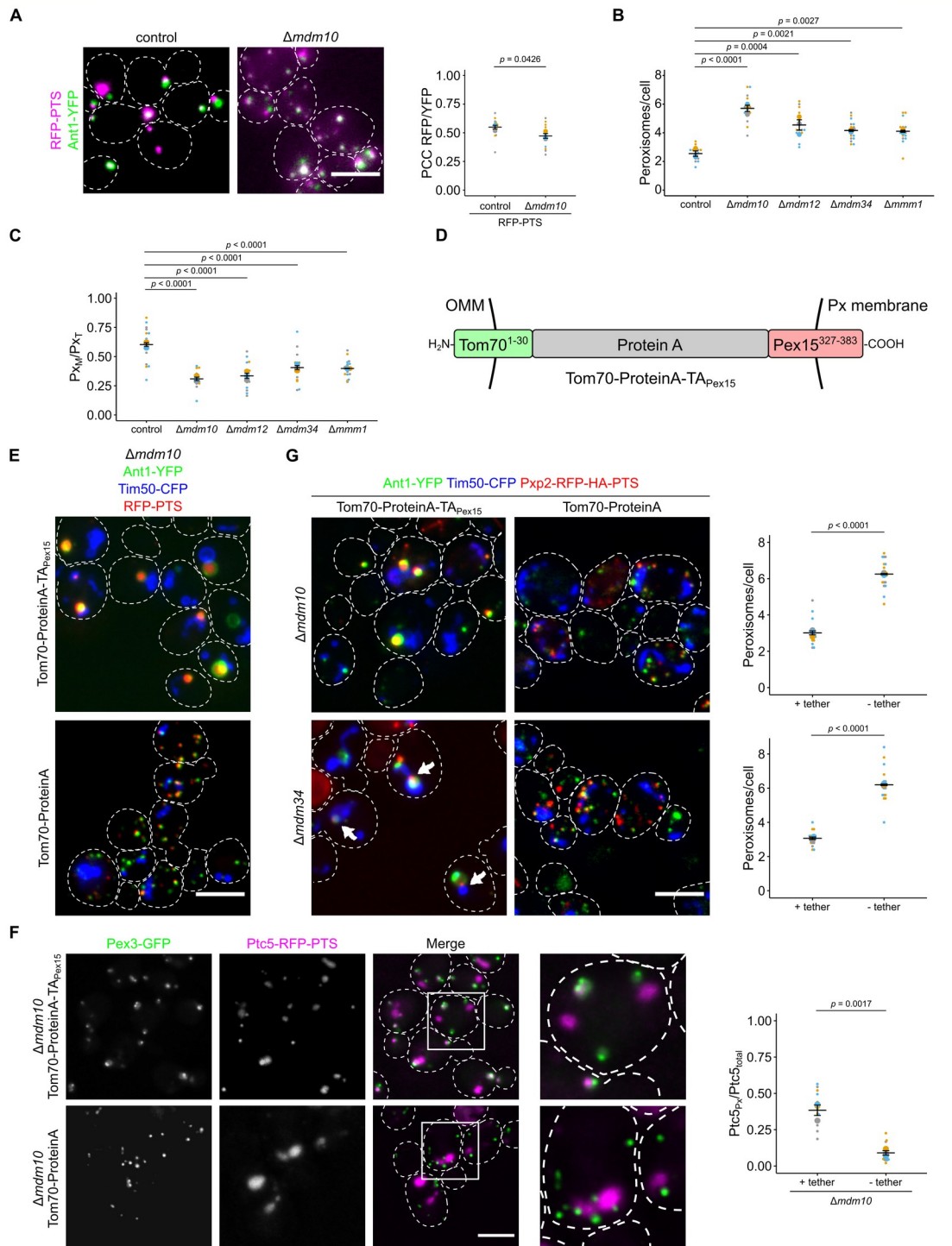

**Fig 7. Peroxisome function and proximity to mitochondria is affected in ERMES mutants. (A)** Fluorescence microscopic images of control and Δ*mdm10* cells co-expressing the peroxisomal marker RFP-PTS (magenta) and the peroxisomal membrane protein Ant1-YFP (green) (left). Correlation of the RFP-PTS signal and the Ant1-YFP signal was quantified using PCC (right). **(B)** The number of peroxisomes per cell was quantified in indicated strains. **(C)** Quantification of the ratio of peroxisomes in contact with mitochondria (Px$_M$) to the total peroxisome count (Px$_T$) in indicated strains. **(D)** Scheme of the synthetic PerMit tether. OMM, outer mitochondrial membrane; Px, peroxisome; TA, tail-anchor. **(E)** A synthetic tether for peroxisomes and mitochondria (Tom70-ProteinA-TA$_{Pex15}$) can suppress accumulation of small peroxisomes. Fluorescence microscopic images of strains deleted for *MDM10* expressing RFP-PTS (red) and the marker proteins Ant1-YFP (green) and Tim50-CFP (blue) in presence of the tether Tom70-ProteinA-TA$_{Pex15}$ or a control protein. **(F)**

Fluorescence microscopic images of Δ*mdm10* cells expressing Ptc5-RFP-PTS (magenta) and Pex3-GFP (green) together with the tether Tom70-ProteinA-TA$_{Pex15}$ or a control protein. Quantifications show the ratio of peroxisomal versus total Ptc5-RFP-PTS signal. **(G)** Fluorescence microscopic images of strains deleted for *MDM10* expressing Pxp2-RFP-PTS (red) and the marker proteins Ant1-YFP (green) and Tim50-CFP (blue) in presence of the tether Tom70-ProteinA-TA$_{Pex15}$ or a control protein. Arrows indicate Pxp2-RFP-PTS foci located at junctions between mitochondria and peroxisomes in Δ*mdm34* cells. Quantification of the number of peroxisomes per cell of the indicated strains (right). Scale bars represent 5 μm. Quantifications are based on *n* = 3 experiments. Each color represents 1 experiment. Error bars represent SEM. *P*-values were calculated using a two-sided unpaired Student's *t* test. For multiple comparisons, *P*-values were calculated with a one-way ANOVA combined with a Tukey test. Underlying data for quantifications can be found in S1 Data. ERMES, endoplasmic reticulum–mitochondria encounter structure; PCC, Pearson's correlation coefficient; PTS, peroxisome targeting signal; RFP, red fluorescent protein.

peroxisome number (Figs 7E–7G, S14, and S15A). In addition, this tether substantially increased the size of peroxisomes, which allowed distinction of membrane and lumen (Fig 7E).

Next, we analyzed the organellar localization of dually targeted proteins upon expression of the synthetic tether. Peroxisomal targeting of Ptc5-RFP-PTS in Δ*mdm10* cells was partially restored upon expression of the synthetic tether (Fig 7F). This finding strengthens the proposed role of Mdm10 as a regulator of PerMit contacts and suggests that the transit from mitochondria to peroxisomes depends on the proximity of both organelles. Interestingly, we were able to visualize Pxp2-RFP-PTS, but not the soluble peroxisomal marker protein RFP-PTS accumulating at junctions between mitochondria and enlarged peroxisomes upon expression of the synthetic tether (Figs 7G and S15). Hence, dual affinity proteins such as Pxp2 can accumulate at contact sites.

Deletion of *PEX5* and deletion of *MDM10* both resulted in a higher number of small peroxisomes, with Δ*mdm10* having the stronger effect (S16A Fig). The phenotype of the double mutant was more reminiscent to Δ*mdm10* single mutants suggesting a functional connection of both factors (S16A Fig). The stronger phenotype of the Δ*mdm10* deletion strain points to additional functions of ERMES for peroxisome formation beside its role in regulating trafficking of MTS-PTS proteins. These could include enhanced activity of fission proteins such as the dually localized tail-anchored (TA) fission protein Fis1 on peroxisomes [49] in cells lacking ERMES. Additional deletion of *FIS1* in Δ*mdm10* cells partially suppressed peroxisome accumulation (S16B Fig).

Together, our data reveal that ERMES regulates contact site formation between peroxisomes and mitochondria.

## The TA-protein Pex15 contributes to ER-peroxisome proximity

Next, we asked if dual targeting dependent tethering occurs at another pair of organelles. We have shown that a synthetic PTS1 protein localized to the ER membrane tethers peroxisomes to this organelle [11]; however, we did not identify such proteins encoded in the yeast genome. We reasoned that in analogy to dual targeting of MTS-PTS proteins, dually targeted membrane proteins may contribute to contact site formation. Many of the proteins identified in a previous screen on PerMit tethers code for such membrane proteins, e.g., Gem1, Fis1, or Fzo1 [17]. Peroxisomal membrane proteins localize to the ER or mitochondria in peroxisome-deficient strains lacking the chaperone Pex19 or the membrane protein Pex3 [50,51]. One such protein is the TA protein Pex15 (Pex26 in mammals) which may be sorted via the ER as well as directly imported into peroxisomes via Pex19 [52–57]. Pex15 can be targeted and translocated into the ER via the GET complex [52].

We expressed the reporter protein ProteinA-(PA)-GFP-Pex15g [53] containing a C-terminal glycosylation tag (g) under control of the *MET25* promoter in the presence of methionine. The chimeric reporter protein predominantly colocalized with RFP-PTS containing foci in

control cells or upon deletion of *GET3* and largely in the ER in *Δpex19* cells (Fig 8A and 8B). This behavior is similar to several dual affinity PTS1 cargo in *Δpex5* and *Δmdm10* mutants, respectively. Cells depleted for members of the GET complex showed an accumulation of small aberrant fast-moving peroxisomes—another analogy to the aforementioned mutants (Fig 8A and 8C and S2 Movie). Thus, the GET pathway is not critical to target PA-GFP-Pex15g to peroxisomes but seems to be involved in the maintenance of regular peroxisomes.

Next, we analyzed the glycosylation status of the reporter protein (Fig 8D). In control cells, we found glycosylated and non-glycosylated PA-GFP-Pex15g indicating dual targeting. In cells depleted for components of the GET complex the fraction of glycosylated protein was reduced, while in *Δpex19* cells PA-GFP-Pex15g was almost entirely glycosylated. This result together with our localization data suggests 2 parallel pathways for Pex15 targeting to peroxisomes. Both direct targeting and transit of Pex15 through the ER appear to be possible.

How does Pex15 leave the ER? This may happen via emergence of vesicular carriers as previously observed by in vitro assays [53]. Alternatively, direct dislocation and subsequent insertion into peroxisomes may occur. The ATPase Spf1 is involved in dislocation of TA proteins from the ER and in peroxisome biogenesis and function [16,58–60]. To test whether this protein is required for sorting of Pex15 from the ER to peroxisomes, we analyzed the localization of Pex15 fused to the bright fluorescent protein mNeonGreen in *Δspf1* cells. We observed significant retention of mNeonGreen-Pex15 in the ER upon deletion of *SPF1* suggesting that Spf1 supports exit from the ER (Figs 8E, 8F, and S17A). Next, we created a mutant lacking Get2 and Spf1. In this double mutant, mNeonGreen-Pex15 predominantly localized in peroxisomes (Figs 8G and S17B). This result confirms the suggested pathway. Pex15 transits through the ER via the GET pathway and Spf1.

As peroxisomal localization of PA-GFP-Pex15g was not abolished by depletion of GET complex members (Fig 8A), accumulation of small fast-moving peroxisomes does probably not result from lack of Pex15 on peroxisomes. To test if ER localization of Pex15 contributes to the maintenance of regular peroxisomes, we overexpressed GFP-Pex15g via a galactose inducible promoter and observed suppression of the *Δget2* and *Δget3* phenotypes (Figs 8H and S18). Prolonged overexpression of PA-GFP-Pex15g resulted in peroxisomes attached to the ER (S19A Fig) resembling peroxisomes attached to mitochondria by overexpression of dual affinity MTS-PTS proteins.

Interestingly, derivatives of Pex15 without the glycosylation tag partially colocalized with mitochondria in the absence of a functional GET pathway (S19B Fig) [52,61]. Hence, the glycosylation tag in Pex15 may increase ER targeting or retention. We speculated that mitochondrial localization of Pex15 would increase tethering of peroxisomes to this organelle since its overexpression was found to affect the signal of a PerMit reporter [17]. Therefore, we created a chimeric fusion protein consisting of the tail anchor of the mitochondrial membrane protein Tom22 and the N-terminal part of Pex15. Induced expression of this synthetic protein substantially enhanced PerMit contacts (Figs 8I and S19C) suggesting that proteins with 2 or multiple destinations more generally affect the extent of proximity between organelles dependent on their subcellular distribution or transit route.

## Synthetic genetics interactions reveal redundant modes for tethering

If tethering of peroxisomes to the ER can be supported by peroxisomal proteins sorted via the ER, then this must be a parallel or redundant way to create ER–peroxisome contacts. Pex30 and orthologous proteins from other fungi that contribute to ER–peroxisome tethering, may regulate membrane flux to peroxisomes, and act as a hub for peroxisome biogenesis at the ER [18,19,62–66]. Similar to *Δget2* and *Δmdm10*, *Δpex30* cells are characterized by an elevated

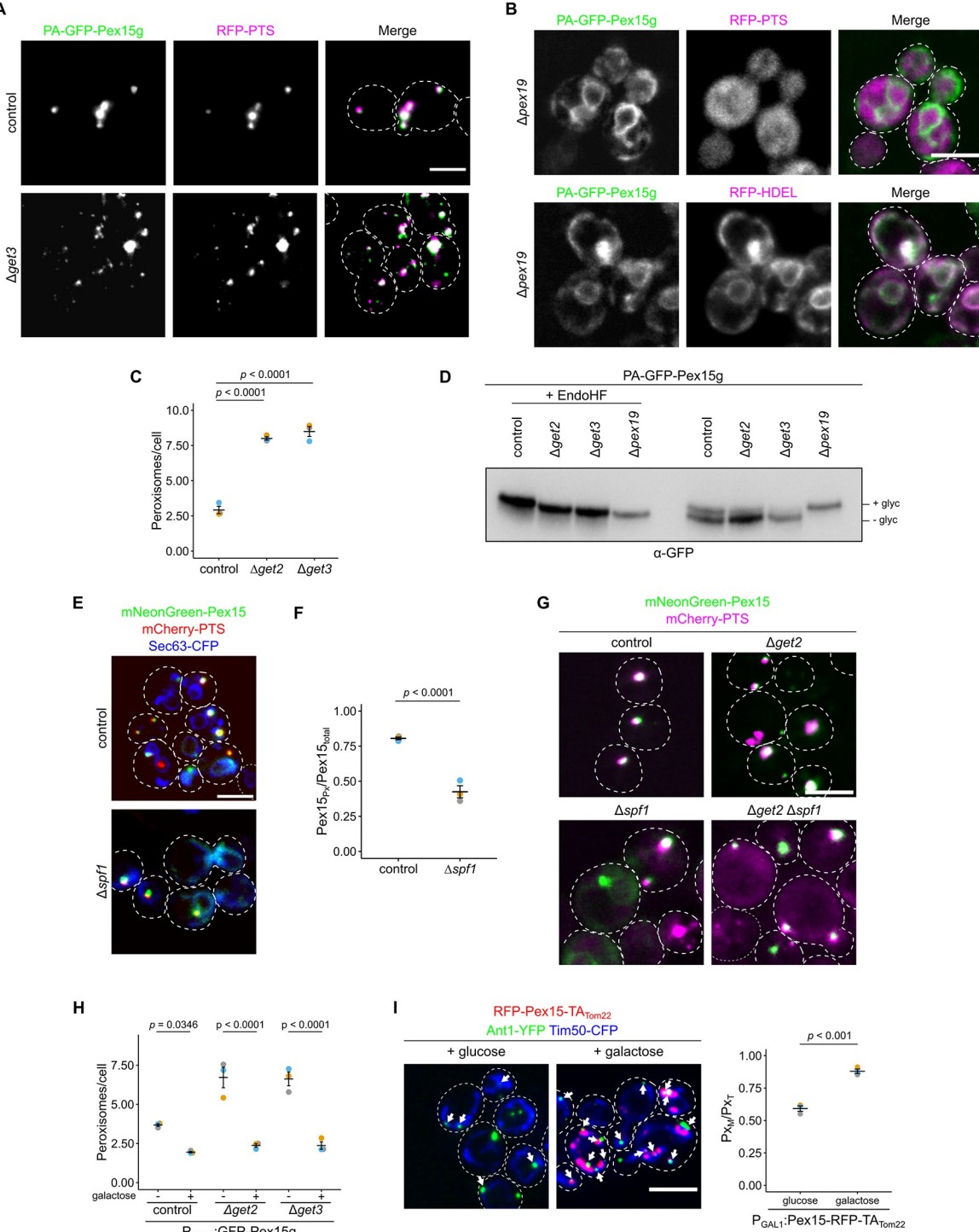

**Fig 8. Dual targeting of Pex15 supports ER-peroxisome contacts. (A)** Fluorescence microscopic images of control and Δ*get3* cells expressing PA-GFP-Pex15g (green) and RFP-PTS (magenta). **(B)** Localization of PA-GFP-Pex15g (green) in the ER in cells lacking the chaperone and targeting factor Pex19. RFP-HDEL (magenta) is a marker protein for the ER; RFP-PTS (magenta) is a marker for peroxisomes. **(C)** Quantification of peroxisome number of indicated strains expressing the peroxisomal marker mCherry-PTS. **(D)** Proteins extracted from indicated strains were subjected to high-resolution SDS-PAGE and immunoblot to visualize glycosylation of PA-GFP-Pex15g. To confirm glycosylation, extracts were treated with endoglycosidase H (EndoHF). **(E)** Fluorescence microscopic images of control and Δ*spf1* cells expressing mNeonGreen-Pex15 (green), mCherry-PTS (magenta), and Sec63-CFP (blue). **(F)** Quantification of the mNeonGreen-Pex15 signal distribution in strains shown in (D). **(G)** Fluorescence microscopic images of control

and indicated mutant cells expressing mNeonGreen-Pex15 (green) and mCherry-PTS (magenta). **(H)** Quantification of peroxisome number of indicated strains either grown in glucose medium or galactose medium for 3 h to induce expression of GFP-Pex15g. **(I)** Induced expression of the chimeric protein RFP-Pex15-TA$_{Tom22}$ (magenta) in cells expressing Ant1-YFP (green) and Tim50-CFP (blue) (left). Quantification of the number of peroxisomes proximal to mitochondria (right). White arrows specify peroxisomes close to mitochondria. Scale bars represent 5 μm. Quantifications are based on $n = 3$ experiments. Each color represents 1 experiment. Error bars represent standard error of the mean. *P*-values were calculated using a two-sided unpaired Student's *t* test. For multiple comparisons, a one-way ANOVA combined with a Tukey test was performed. Underlying data for quantifications can be found in S1 Data. ER, endoplasmic reticulum; PTS, peroxisome targeting signal; RFP, red fluorescent protein.

number of smaller, highly mobile peroxisomes (S2 Movie) [19]. We reasoned that phenotypic analysis of double mutants may help to clarify the relation of factors for sorting and tethering at different organelles. Previously, a negative genetic interaction between *Δget2* and *Δpex30* was shown [67,68]. We observed substantial cytosolic mistargeting of mCherry-PTS, reduced cell growth and altered mitochondrial morphology in *Δget2Δpex30* double mutants (Figs 9A and S20).

We propose that peroxisome formation requires ER-tethering facilitated by parallel mechanisms—Pex30 on the one hand and protein targeting via the GET complex on the other.

Finally, we addressed the phenotypes of cells lacking sorting and tethering machinery at both organelles. Further synthetic genetic interactions were detected in comprehensive interaction maps [67,68], e.g., *Δmdm10Δpex30* (positive genetic or buffering) and *Δmdm12Δget2* (negative genetic or synthetic lethal). In a *Δmdm10Δpex30* strain, both the mitochondrial morphology phenotype and the growth defect of *Δmdm10* single mutants were partially suppressed showing that absence of Pex30 counteracts absence of Mdm10 (Figs 9B and S20C). Mitochondrial morphology and function seem to benefit if a major hub for several other organelles [64–66] is missing at the ER. In addition to substantial cytosolic localization of mCherry-PTS, *Δget2Δmdm10* cells contained an increased number of small peroxisomes compared to each of the single mutants (Fig 9C and 9D). This reveals that ERMES and GET complexes are involved in independent processes with respect to peroxisome formation.

## Discussion

We have found that distinct proteins with targeting signals for 2 organelles can affect proximity of these organelles. This conclusion is supported by the notion that different types of dual affinity proteins can act as contact-inducing proteins (Fig 10), the phenotypic similarity of *Δget2*, *Δmdm10*, and *Δpex5* strains with respect to peroxisome number, morphology and movement, and the phenotypes of several double mutants. Although dual affinity proteins are a challenge for maintaining organelle identity [69,70], they are ideally suited to support organelle interactions by binding to targeting factors and membrane-bound translocation machinery of different organelles. Dually targeted proteins appear to concentrate in regions of organelle contact, which may coincide with regions of reduced identity.

Localization of peroxisomal membrane proteins to mitochondria has been observed in mutants deficient in Pex19, components of the GET complex or the AAA-ATPase Msp1 [51,52,61]. Mitochondrial targeting of overexpressed Pex15 is observed upon depletion of components of the GET complex but not in control cells (S19B Fig) [52], thus sorting via the ER probably represents a major avenue of transit.

If ERMES has a direct function in peroxisome tethering or if it is able to directly translocate lipids to peroxisomes remains elusive. The latter was recently shown for ERMES at ER–mitochondria junctions [71]. We show here that depletion of ERMES has a huge impact on mitochondrial protein import (Figs 6 and S10A). This phenotype is likely to affect tethering of mitochondria to different organelles including the ER.

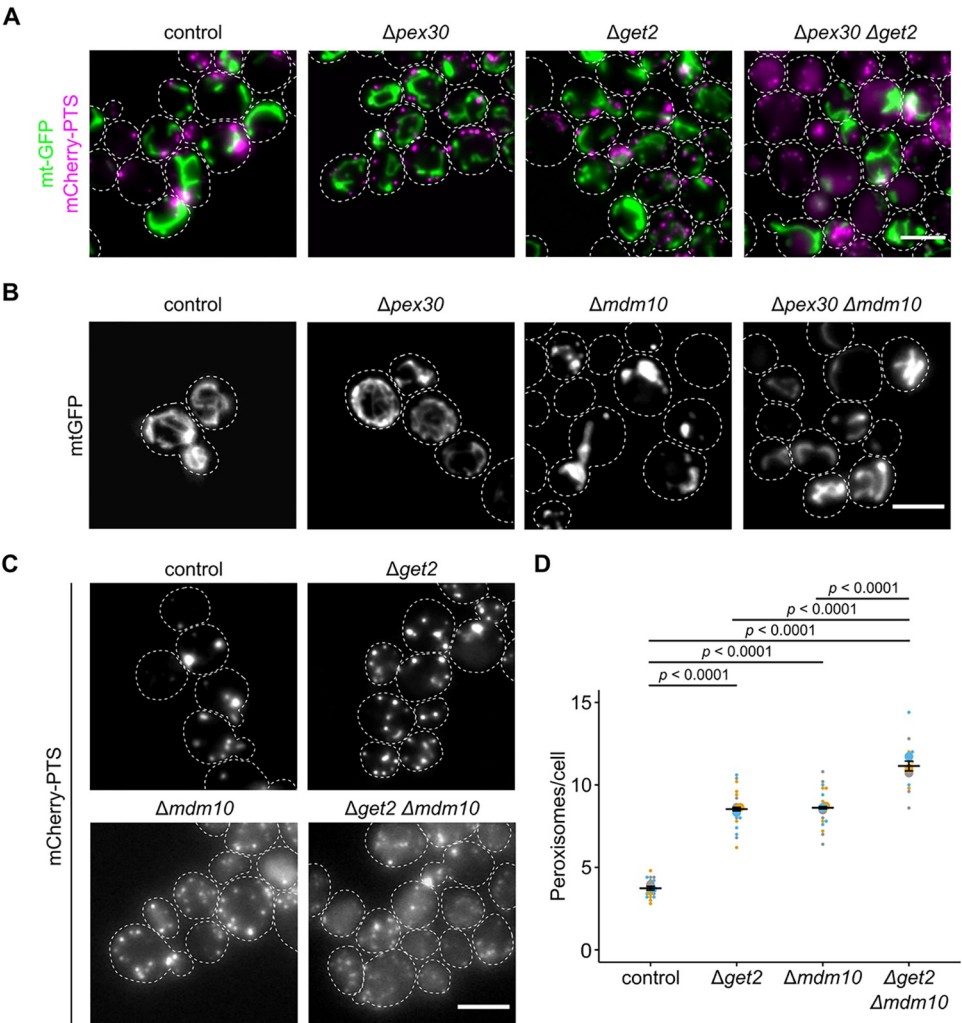

**Fig 9. Synthetic genetic interactions reveal redundant modes of tethering. (A)** Fluorescence microscopic images of indicated strains expressing the peroxisomal marker mCherry-PTS (magenta) and the mitochondrial marker mt-GFP (green). **(B)** Fluorescence microscopic images of indicated strains expressing the mitochondrial marker mt-GFP. **(C)** Fluorescence microscopic images of indicated strains expressing the peroxisomal marker mCherry-PTS. **(D)** Quantification of the peroxisome number in indicated strains. Scale bars represent 5 μm. Quantifications are based on $n = 3$ experiments. Each color represents 1 experiment. Error bars represent standard error of the mean. $P$-values were calculated using a two-sided unpaired Student's $t$ test. Underlying data for quantifications can be found in S1 Data. PTS, peroxisome targeting signal.

Other previous observations are consistent with our concept of dual targeting-induced tethering. Increased levels of various dual affinity proteins enhanced PerMit contacts [17]. Ant1 and Pex11 enrich in mitochondria of mutants lacking components required for peroxisomal membrane protein import [16,18]. While many peroxisomal membrane proteins can target peroxisomes without transitioning through the ER [72], several peroxisomal membrane proteins have evolved to be synthesized in vicinity to the ER and may translocate from it [73]. One outcome of this detour may be increased tethering. An example is the integral membrane protein Pex3 that can either transit through the ER or directly insert into peroxisomes [74,75]. Remarkably, artificial targeting of Pex3 to mitochondria is compatible with peroxisome biogenesis in yeast [76].

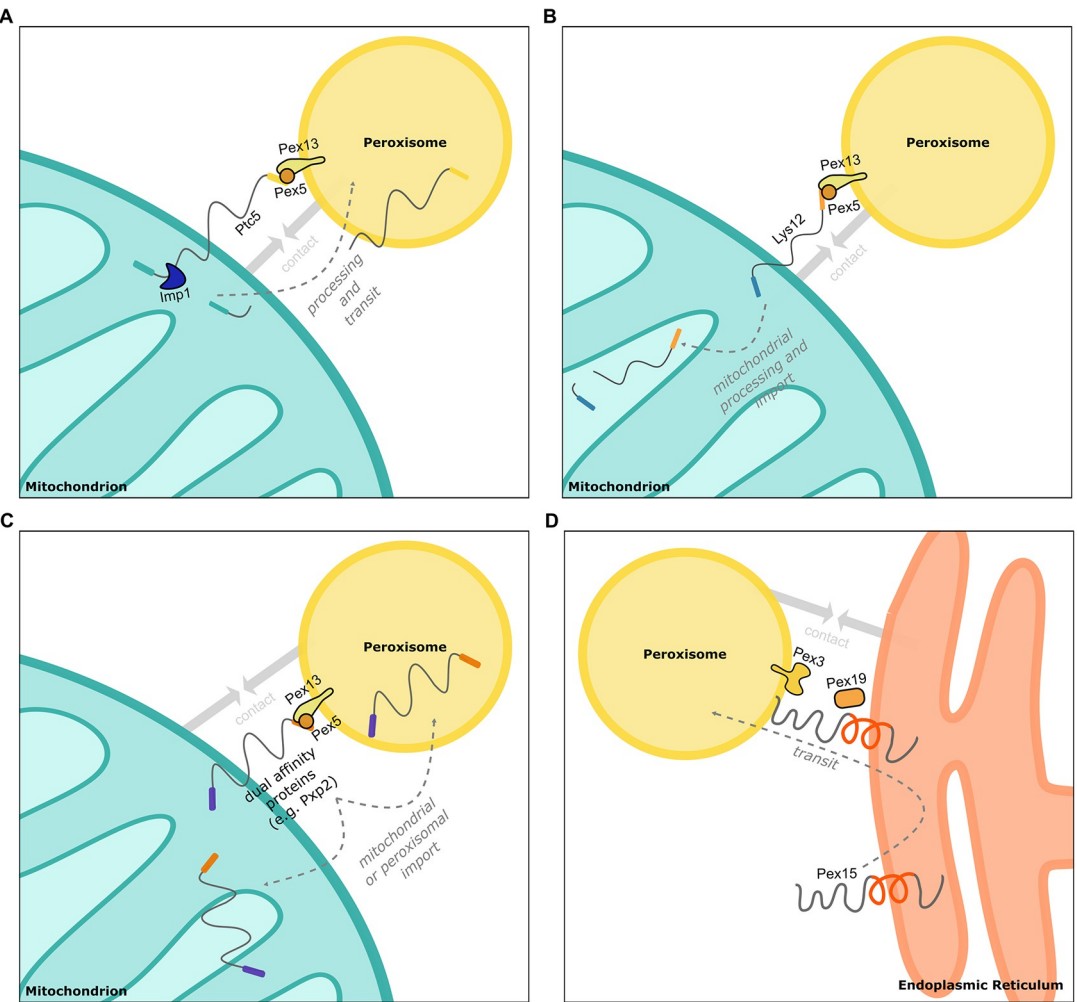

**Fig 10. Examples for tethering via dual affinity proteins. (A)** Ptc5 transits mitochondria to peroxisomes and supports tethering. After processing by the peptidase Imp1, Ptc5 is imported into peroxisomes or in the mitochondrial intermembrane space [11]. **(B)** Lys12 contains a PTS1 that affects proximity between peroxisomes and mitochondria. Translocation to peroxisomes was not observed. **(C)** Other dual affinity proteins such as Pxp2 contribute to the formation of contacts between peroxisomes and mitochondria and are dually targeted. If transit from one to the other organelle occurs is unknown. **(D)** Pex15 can directly target peroxisomes or transit through the ER. ER localization promotes tethering. ER, endoplasmic reticulum; PTS, peroxisome targeting signal.

The decision to which organelle a peroxisome will preferentially attach can be influenced by the metabolic state of the cell [17]. In oleic acid medium proteins such as Tes1 and Cat2 are highly induced which may account for the observation of more frequent connections to mitochondria [17,77]. We observed an accumulation of *U. maydis* Ptc5 in foci at the interface of mitochondria and peroxisomes upon fatty acid treatment (Fig 1). In addition, we find that lysine deprivation increases PerMit connections and that this requires a functional PTS1 in the mitochondrial protein Lys12 (Fig 4).

Does this mode of targeting and tethering only occur for peroxisomal proteins? In mammalian cells, ER localization of the bona fide mitochondrial protein Mitofusion 2 tethers mitochondria and the ER [78]. Interestingly, ER-specific isoforms of Mitofusin2 can be generated via alternative splicing. One of these variants acts as a tether for mitochondria at the site of the ER [79]. In yeast, the recently identified tether for mitochondria and the nuclear envelope

Cnm1 shows hallmarks of a dual affinity protein comprising a targeting signal for the ER and for mitochondria at the N- and C-termini, respectively [80]. The discovery of an ER surface retrieval pathway for mitochondrial membrane proteins [81] exemplifies a rerouting mechanism with some similarity to Ptc5 and Pex15, but if this ER transit of mitochondrial proteins affects ER–mitochondria tethering is currently unknown. The chaperone Djp1 critical in this pathway could function similar to Pex5 or Pex19—the latter has an unexpected role in sorting of mitochondrial TA proteins [82]. We detected Δ*djp1* among the mutants that decrease import of Ptc5-RFP-PTS to peroxisomes (Fig 5). It was shown previously that loss of Djp1 affects peroxisomal protein import [83]. The GET complex mediates import of "peroxisomal and mitochondrial" proteins into the ER [52,84]. Several studies uncovered an overlap of factors for peroxisome and mitochondrial biogenesis in mammalian cells [85–87]. Accordingly, shared machinery controls protein targeting and biogenesis of both organelles and may thus contribute to tethering as well.

We conclude that dually targeted cargo includes a diverse and unexpected group of tethers, which are likely to maintain contact as long as they remain accessible for targeting factors at partner organelles. Coupling of protein and membrane trafficking is a common principle in the secretory pathway [88] and it might also occur for peroxisomes at different contact sites. How dually targeted proteins and their rerouting affect the flux of molecules other than proteins, e.g., membrane lipids remains a topic for future research.

## Methods

### Yeast strains, plasmids, and oligonucleotides

All *S. cerevisiae* strains used in this study have the genetic background of BY4741 or BY4742 [89] and are listed in S4 Table. Yeast cells were transformed as previously described using Li-acetate, PEG, and ssDNA [90]. Gene replacement or endogenous tagging was carried out using PCR-amplification of cassettes based on an established toolbox system. The following plasmids were used for PCR amplification for endogenous tagging in yeast: pKS133, pKS134, pUG72, pHyg-AID*-6HA, pYM25, pCR125, and pYM-N22 [32,91–93]. Plasmids were generated by Gibson Assembly, recombinational cloning or classic molecular cloning [94–96]. For *U. maydis*, plasmids were derived from pETEF-ALA6-MMXN [97]. Plasmids used or generated and respective oligonucleotides are listed in S4 Table [11,53,89,98–101]. Stable integration of overexpression constructs was achieved via homologous recombination into the LEU2 locus using primers overlapping regions flanking the deletion of the open reading frame.

### Yeast media and growth conditions

Strains were grown in either YPD (1% yeast extract, 2% peptone, 2% glucose) medium or synthetic medium containing 2% glucose, 0.67% yeast nitrogen base without amino acid, 0.5% ammonium sulfate and complementary supplement mixtures (CSMs) or drop-out mixtures lacking appropriate amino acids when experiments required selection or lysine deprivation. For galactose induction of RFP-Pex15g, cells were grown overnight in synthetic medium containing 2% raffinose, 0.1% glucose, 0.67% yeast nitrogen base without amino acids, 0.5% ammonium sulfate and CSM-His or CSM-His-Leu overnight. Cells were washed once with $H_2O$ and diluted in 2% raffinose, 0.67% yeast nitrogen base without amino acids and CSM-His or CSM-His-Leu. After 2 h, 2% galactose was added and cells were incubated for 3 h. For regulation of Pex5 expression, cells were grown overnight in YP (1% yeast extract, 2% peptone) medium containing 2% raffinose and 0.1% glucose. Cells were washed once with $H_2O$ and diluted to an $OD_{600}$ of 1 in YP containing 2% raffinose. After 2 h, cells were harvested and resuspended in YP medium containing 2% galactose. After 45 min, cells were again harvested

and resuspended in YPD medium. Samples were taken at indicated time points and analyzed either by fluorescence microscopy or SDS-PAGE and immunoblot. For depletion of Pex13, cells were precultured in either YPD or synthetic medium containing 2% glucose, 0.67% yeast nitrogen base without amino acids, 0.5% ammonium sulfate and dropout-mix lacking histidine. Cells were diluted in the appropriate medium to an $OD_{600}$ of 0.5 and incubated at 30˚C for 2 h. The culture was then split into 2 aliquots of equal volume and 1 culture was supplemented with 2 mM indole-3-acetic acid (auxin). Both cultures were incubated at 30˚C and at indicated time points samples were taken and analyzed by either fluorescence microscopy or SDS-PAGE and immunoblot. For growth assays on plates, 5 µl of serial tenfold dilutions of logarithmically growing cells (starting $OD_{600}$ = 1) were spotted onto solid synthetic media either containing 2% glucose or 2% glycerol and were incubated for 2 to 5 days at 30˚C. Cells expressing PA-GFP-Pex15g under control of a methionine regulatable promotor [53] were grown in the presence of 200 µm methionine or 50 µm methionine to achieve substantial overexpression.

## *U. maydis* growth and manipulation

*U. maydis* strains generated and used in this study are derivatives of Bub8 and Bub8 mCherry-PTS and listed in S4 Table. *U. maydis* cells were grown at 28˚C in YEPSlight medium or yeast nitrogen base medium (YNB; Difco) supplemented either with 2% glucose and 0.2% ammonium sulfate or 0.1% oleic acid (Roth), 0.1% Tween-40 and 0.2% ammonium sulfate at pH 5.6. For oleic acid induction experiments, cells were incubated for 4 h in respective medium. Cell transformation was performed as previously described [100,101]. DNA was integrated into the *IP*-locus of *U. maydis* cells [102]. Carboxin (2 µg/ml) was used for selection. Genomic DNA was extracted as previously described [103].

## High-content screening

To create collections of haploid strains, we constructed a Synthetic Genetic Array (SGA) compatible query strain co-expressing the peroxisomal marker Pex3-GFP (endogenous) together with Ptc5-RFP-PTS (overexpression). Using automated mating approaches the query strain was crossed with deletion and DAmP arrayed collections [40,42,43]. Yeast manipulations in high-density format were performed on a RoToR bench top colony arrayer (Singer Instruments). In short: mating was performed on rich medium plates, selection for diploid cells was performed on SD monosodium glutamate (MSG) plates containing geneticin (Formedium, 500 µg ml$^{-1}$), nourseothricin (WERNER BioAgents "ClonNat," 200 µg ml$^{-1}$) and hygromycin B (Formedium, 500 µg ml$^{-1}$). Sporulation was induced by transferring cells to nitrogen starvation media plates for 7 d. Haploid cells containing the desired mutations were selected by transferring cells to SD MSG plates containing geneticin, neurseothricin, and hygromycin B, alongside the toxic amino acid derivatives canavanine and thialysine (Sigma-Aldrich) to select against remaining diploids, and lacking leucine to select for spores with an alpha mating type.

The collections were visualized using an automated microscopy setup. In brief, cells were transferred from agar plates into 384-well polystyrene plates for growth in liquid media using the RoToR arrayer. Liquid cultures were grown in a LiCONiC incubator overnight at 30˚C in SD medium (6.7 g l$^{-1}$ yeast nitrogen base with ammonium sulfate and 2% glucose) supplemented with complete amino acids. A JANUS liquid handler (PerkinElmer) connected to the incubator was used to dilute the strains to an $OD_{600}$ of approximately 0.2. Plates were incubated at 30˚C for 4 h in SD medium. The cultures in the plates were then transferred by the liquid handler into glass-bottom 384-well microscope plates (Matrical Bioscience) coated with concanavalin A (Sigma-Aldrich). After 20 min, wells were washed twice with SD-Riboflavin complete medium to remove non-adherent cells and to obtain a cell monolayer. The plates

were then transferred to a ScanR automated inverted fluorescent microscope system (Olympus) using a robotic swap arm (Hamilton). Images of cells in the 384-well plates were recorded in SD-riboflavin at 24°C using a ×60 air lens (NA 0.9) with an ORCA-ER charge-coupled device camera (Hamamatsu). Images were acquired in 2 channels: GFP (excitation filter 490/20 nm, emission filter 535/50 nm) and mCherry (excitation filter 572/35 nm, emission filter 632/60 nm). After acquisition, images were manually reviewed using the ScanR analysis program and ImageJ.

## Preparation of post-nuclear supernatants and differential centrifugation

For preparation of crude organelles, 3 to 6 liter cultures were grown in synthetic medium containing appropriate amino acids for selection to logarithmic phase ($OD_{600}$ = 0.8–1) at 30°C. Cells were harvested and suspended in 25 ml 100 mM Tris/HCl buffer (pH 9.4) per liter of starting culture containing 10 mM DTT. The suspension was incubated for 10 min at room temperature and centrifuged at $600 \times g$ for 5 min. Cell pellets were suspended in 25 ml lyticase buffer (0.7 M sorbitol, 0.75xYP, 0.5% glucose, 10 mM Hepes/OH, 1 mM DTT (pH 7.4)) per liter of starting culture and lyticase ($10^5$ units per liter starting culture) was added. Suspensions were incubated for 30 to 45 min at 30°C and the efficiency of spheroplast formation was determined by measuring the decline of $OD_{600}$ after suspension of samples in $H_2O$. Spheroplasts were washed 3 times with 25 ml 2xJR buffer (0.4 M sorbitol, 100 mM KOAc, 40 mM Hepes/OH (pH 7.4), 4 mM EDTA, 2 mM DTT) per liter starting culture, suspended in 5 ml 2xJR (containing a protease inhibitor cocktail: 1 mM 4-aminobenzamidinedihydrochloride, 1 μg/ml aprotinin, 1 μg/ml leupeptin, 1 mM phenylmethylsulfonyl fluoride, 10 μg/ml N-tosyl-L-phenylalanine chloromethyl ketone, and 1 μg/ml pepstatin) per liter starting culture and frozen at −80°C. Cells were thawed in iced water and disrupted by 20 strokes with a Potter-Elvehjem homogenizer. Nuclei were removed from homogenates by centrifugation 2 times at $600 \times g$. For differential centrifugation, 1 mg post-nuclear supernatant (PNS) fraction was used. PNS (100 μl) fractions were centrifuged at $13k \times g$ for 5 min. The resulting supernatant was centrifuged at $100k \times g$ for 20 min. All fractions were analyzed by SDS-PAGE and immunoblot in amounts representing the initial volume.

## Density gradient centrifugation

PNSs for density gradient centrifugation were prepared using a modified protocol from Cramer and colleagues (steps 1–16) [104]. Cells were precultured in 500 ml YP medium supplemented with 0.1% glucose at 30°C overnight. Cells were harvested at $6,000 \times g$ for 6 min, washed once with 30 ml sterile water, and resuspended in 500 ml YNBO medium. After incubation at 30°C for 16 h, cells were harvested as described above, washed twice with 30 ml sterile water, and incubated in 15 ml DTT buffer (100 mM Tris, 10 mM DTT (pH 7.4)) for 30 min at 30°C with gentle agitation. Cells were sedimented by centrifugation at $600 \times g$, washed 3 times with 15 ml sorbitol buffer (20 mM HEPES, 1.2 M sorbitol), and incubated with 15 ml sorbitol buffer containing 20 mg Zymolyase 100 T (Roth) at 30°C with gentle agitation. Digestion of yeast cell walls was monitored photometrically at OD600 and digestion was stopped when at least half of the cells lysed upon addition of water. From this point, all steps were carried out on ice or at 4°C. Spheroplasts were gently washed 3 times with 15 ml sorbitol buffer, resuspended in 15 ml lysis buffer (5 mM MES, 0.5 mM EDTA, 1 mM KCl, 0.6 M sorbitol, 1 mM 4-aminobenzamidine-dihydrochloride, 1 μg/ml aprotinin, 1 μg/ml leupeptin, 1 mM phenylmethylsulfonyl fluoride, 10 μg/ml N-tosyl-L-phenylalanine chloromethyl ketone, and 1 μg/ml pepstatin), and frozen at −80°C overnight. Spheroplasts were thawed on ice and homogenized using a Potter-Elvehjem homogenizer ($2 \times 12$ strokes). Nuclei and cell debris were removed by

2 subsequent centrifugations at 1,600 × g for 10 min. Subsequently, the PNS was diluted to an OD600 of 1, aliquoted and frozen at −80˚C. A total of 200 µl of PNS without cytosol were loaded onto a Nycodenz density gradient consisting of 333 µl 20%, 666 µl 25%, 666 µl 30%, and 333 µl 35% Nycodenz in gradient buffer A (5 mM MES, 1 mM EDTA, 1 mM KCl, and 0.1% (v/v) ethanol). Gradients were centrifuged in a Beckman L7-65 ultracentrifuge equipped with a Sorvall TST 60.4 rotor at 100,000 × g (35,000 rpm) for 90 min at 4˚C. Twelve 183 µl fractions were collected from the top of the gradient and used for SDS-PAGE and immunoblot.

## Epifluorescence microscopy

A total of 200 µl of hot 1.5% agarose melted in water was used to create a thin agarose cushion on a 76 × 26 mm microscope slide (Roth, Karlsruhe, Germany). Cells were washed with water, concentrated 10-fold, and 3 µl aliquots were spotted onto the middle of the agarose pad and covered with an 18 × 18 mm coverslip (Roth, Karlsruhe, Germany). Microscopy was performed on an Axiovert 200 M inverse microscope (Zeiss) equipped with a 1394 ORCA ERA CCD camera (Hamamatsu Photonics), filter sets for cyanGFP, enhanced GFP (EGFP), yellow fluorescent protein (YFP), and rhodamine (Chroma Technology, Bellows Falls, Vermont, United States of America), and a Zeiss 63× Plan Apochromat oil lens (NA 1.4). Single-plane bright field or phase contrast images and z-stacks of the cells (0.5 µm z-spacing) in the appropriate fluorescence channels were recorded, using the image acquisition software Volocity 5.3 (Perkin-Elmer). Movies represent a time of 40 s. Images were processed and evaluated in ImageJ [105]. Alternatively, images were taken on an inverted Zeiss Axioobserver fluorescence microscope and analyzed with Metamorph (Molecular Devices). For protein localization analysis, z-projections of deconvolved image stacks of the fluorescent channels were used. Deconvolution was performed on the z-stacks by the ImageJ plugin DeconvolutionLab with 25 iterations of the Richardson–Lucy algorithm [106]. Pearson's correlation coefficients were determined with Volocity 5.3.

## Automated time lapse microscopy

Exponentially growing liquid cultures were spotted on 1.5% agarose pads containing synthetic media. Images were taken with a Zeiss Axio Observer. Z1 microscope equipped with a ×100/1.46 Oil DIC objective and a pco.edge 4.2 sCMOS camera (PCO). An X-Cite 120PC metal halide light source (EXFO, Canada) and ET-YFP or ET-TexasRed filter cubes (Chroma, USA) were used for fluorescence detection. Images were recorded with VisiView 3.3.0.6 (Visitron Systems) and processed with Metamorph 7.7.5 (Molecular Devices) and Adobe Illustrator CS6 (Adobe Systems). Images were taken every hour and cells were incubated at 30˚C.

## Structured illumination microscopy

Approximately 5 µl of logarithmically growing cells were spotted on glass coverslips and covered with a 1% YNB agarose pad. SIM images were acquired by a Zeiss ELYRA PS.1 microscope with an ANDOR iXon 987 EMCCD camera (exposure time: 50 ms, gain: 5) using an alpha Plan-Apochromat 100×/1.46 Oil DIC M27 Elyra objective with 488 nm (HR Diode 488–200, 3%) and 564 nm (HR DPSS 561–200, 5%) laser lines as excitation sources. A total of 15 images (3 rotations, 5 phases) were collected per plane. The software ZEN (Zeiss) was used for processing the SIM reconstructions.

## Transmission electron microscopy and immunogold labeling

For visualization of *U. maydis* cells via TEM, 50 ml logarithmically growing cells were harvested and frozen under high-pressure (Wohlwend HPF Compact 02). After subsequent freeze

substitution (using acetone, containing 0.25% osmium tetroxide, 0.2% uranyl acetate, and 5% water) (Leica AFS2), cells were embedded in Epon812 substitute resin (Fluka). Embedded cells were sectioned to 100 nm thin sections (Leica EM UC7 RT), which were used for immunolabeling with anti-mCherry (A121603; antibodies.com; dilution 1:100) or anti-GFP (600-101-215; Rockland; dilution 1:100). As a secondary antibody rabbit-anti-goat antibody coupled to ultra-small gold particles was used (Aurion, dilution 1:100). Subsequently, a silver enhancement procedure was performed and sections were post-stained with 2% uranyl acetate and 0.5% lead citrate. Analysis of the samples was conducted with a JEOL JEM2100 TEM equipped with a fast-scan 2k CCD TVIPS (Gauting, Germany) F214 camera. Images were processed with ImageJ [105]. For quantifications, 3 samples were characterized for each strain. At least 25 peroxisomes per sample were analyzed.

### Immunoblotting and antibodies

Small amounts of protein were extracted as described by Kushnirov (2000) using a modified sample buffer (50 mM Tris-HCl (pH 6.8), 2% SDS, 6% glycerol, 0.025% bromophenol blue, and 50 mM dithiothreitol) [107]. In brief, 1 $OD_{600}$ of yeast cells were centrifuged at $13,000 \times g$ for 1 min and incubated with 300 μl 0.2 M NaOH for 5 min at room temperature. Cells were centrifuged, resuspended in 50 μl sample buffer, incubated for 5 min at 95°C, centrifuged again, and the supernatant was transferred to a new reaction tube. Proteins were incubated for 5 min at 95°C and rotated at 750 rpm prior to loading. SDS-PAGE was performed with self-cast or Midi-protean TGX precast gels (BioRad), PageRuler Prestained protein ladder (Thermo Fisher) as protein standard and a BioRad Mini- or Midi-Protean cell. Proteins were blotted on PVDF membranes in a BioRad Mini Trans-Blot cell or Criterion Blotter at 30 V for 16 h at 4°C or at 70 V for 1 h at 4°C. Membranes were blocked for 30 min in TBST containing 5% nonfat dry milk and incubated in primary antibody containing 0.02% sodium azide with gentle agitation for 2 h at room temperature or overnight at 4°C. After removal of the antibody solution, membranes were washed 5 times with TBST for 5 min, incubated with HRP-conjugated secondary antibody for 45 min at room temperature, washed 5 times with TBST for 5 min, and then developed using either Pierce ECL Immunoblotting Substrate (Thermo Fisher) or SuperSignal West Pico PLUS Chemiluminescent Substrate (Thermo Fisher). The following antibodies were used in this study: anti-GFP (1:2,000; TP401, Torrey Pines Biolabs), anti-HA (1:2,000; ab1302275, Abcam), anti-tagRFP (1:1,000; AB233, Evrogen), anti-Por1 serum (1:2,000; kindly provided by Roland Lill, Marburg), anti-Pex5 serum (1:10.000; kindly provided by Ralf Erdmann, Bochum), anti-Kar2 serum (1:5,000), anti-Pgk1 (1:2,000; 22C5D8, Thermo Fisher), anti-mCherry (1:1,000; TA150125, Thermo Fisher), anti-Protein A (1:2,000; ab19483, Abcam), goat anti-mouse IgG-HRP (1:10.000–1:50.000; 31430, Thermo Fisher), goat anti-rabbit IgG-HRP (1:10.000–1:50.000; 31460, Thermo Fisher). Uncropped Scans are depicted in S1 Raw Images.

### EndoHF treatment

Protein extract samples (5 μl) were subjected to EndoHf (New England Biolabs) treatment according to manufactures instructions. EndoHf (4 μl, 4,000 units) was used per reaction. To resolve different species of glycosylated and non-glycosylated proteins, we used 7.5% Tris-glycine midi-gels (Bio-Rad).

### Statistics and reproducibility

Microscopic data was collected from 3 independent *S. cerevisiae* cultures. Five images per culture were quantified. Pearson's correlation coefficient was calculated with Volocity 5.5.2. Superplots [108] and Student's *t* tests were computed using RStudio 1.2.1335 with R 3.6.0.

Blots are structured as follows: center line, mean; error bars, standard error of the mean; big circles, mean of experiments; and small circles, data points of experiments. *P*-values were calculated using an unpaired, two-sided Student's *t* test. For data which contain multiple comparisons, a one-way ANOVA combined with a Tukey's post-test was performed to assess significance of the differences. For calculation of Pearson's correlation coefficients, all analyzed images contained 10 or more cells. For quantification of contacts between mitochondria and peroxisomes, 5 images per strain with at least 5 cells from 3 independent experiments were analyzed. Peroxisomes were counted as mitochondria associated if no black pixels were detected between fluorescent signals of respective marker proteins. Inspection was carried out without knowledge of the genotypes. For quantifications shown in Figs 7F and 8E, the intensity of peroxisomal signal versus total signal were measured using ImageJ. All experiments were at least repeated 3 times with similar results.

## Supporting information

**S1 Fig. Um_Ptc5 accumulates at PerMit contacts upon addition of oleic acid. (A)** Fluorescence microscopic images of cells expressing Um_Ptc5-GFP or Um_Ptc5-GFP-PTS (green) under control of the constitutive *Otef*-promoter together with the peroxisomal marker mCherry-PTS (magenta). **(B)** Structured illumination microscopic (SIM) images of strains shown in (A). **(C)** Fluorescence microscopic images of cells expressing an internally GFP-tagged derivative of Um_Ptc5 (green) under control its endogenous promoter together with the peroxisomal marker mCherry-PTS (magenta) following incubation with indicated carbon sources. **(D)** Structured illumination microscopic (SIM) images of strains shown in (C) grown under the same conditions. **(E)** Fluorescence microscopic images of cells expressing a C-terminally GFP-tagged derivative preserving the PTS of Um_Ptc5 (green) at the endogenous locus together with the peroxisomal marker mCherry-PTS (magenta) following incubation with indicated carbon sources. Scale bars: 5 μm.
(TIF)

**S2 Fig. Dually localized proteins induce PerMit contacts upon overexpression in yeast.**
Tes1-RFP-HA-PTS (A; red) or Mss2-RFP-HA-PTS (B; red) were co-expressed with the peroxisomal membrane protein Ant1-YFP (green) and mitochondrial inner membrane protein Tim50-CFP (blue) in indicated strains (left). Subcellular localization was determined by fluorescence microscopy. Scale bar represents 5 μm. White arrows indicate peroxisomes proximal to mitochondria. Quantification of correlation between Tes1-RFP-HA-PTS (top right) signal or Mss2-RFP-HA-PTS (bottom right) signal and Ant1-YFP signal in indicated strains. PCC refers to Pearson's correlation coefficient. Quantifications are based on *n* = 3 experiments. Each color represents 1 experiment. Error bars represent standard error of the mean. *P*-values were calculated using a two-sided unpaired Student's *t* test. (C) Subcellular localization of endogenously tagged variants of dual affinity proteins was determined by density gradient centrifugation. Twelve fractions, collected from the top of the gradient, were analyzed by SDS-PAGE and immunoblot. Ant1-YFP is a peroxisomal membrane protein and Por1 is localized in the mitochondrial outer membrane. (D). Yeast cells expressing N-terminally tagged Pxp2 together with Ant1-YFP and Tim50-CFP were analyzed by fluorescence microscopy. The progenitor strain without RFP-Pxp2 served as control. White arrows denote peroxisomes overlapping mitochondria. Underlying data for quantifications can be found in S1 Data.
(TIF)

**S3 Fig. Depletion of Pex5 reduces PerMit contacts.** Fluorescence microscopic image from control and Δ*pex5* cells expressing endogenously tagged Pex3-GFP (green) **(A)** or

Pex14-mNeonGreen (green) **(B)** and Tim50-RFP (magenta). White arrows denote peroxisomal signal overlapping with mitochondrial signal. Scale bars 5 μm. **(C)** The number of peroxisomes per cell was quantified in strains of (B) (left). Quantification of the fraction of peroxisomes contacting mitochondria ($Px_M$) in relation to the total peroxisome count ($Px_T$) in strains of (B) (right). Scale bars represent 5 μm. Quantifications are based on $n$ = 3 experiments. Each color represents 1 experiment. Error bars represent SEM. $P$-values were calculated using a two-sided unpaired Student's $t$ test. Underlying data for quantifications can be found in S1 Data.
(TIF)

**S4 Fig. Inhibition of peroxisomal protein import via an auxin inducible degron reduces PerMit contacts. (A)** Fluorescence microscopic images of indicated strains expressing the peroxisomal membrane protein Ant1-YFP (green) and RFP-PTS (red) in the absence (-Auxin) or presence (+Auxin; 4 h) of 2 mM indole-3-acetic acid. **(B and C)** Images are from indicated strains expressing Ant1-YFP (magenta) and Tim50-CFP (green) photographed at indicated time points after addition of 2 mM indole-3-acetic acid (auxin). Arrows indicate peroxisomes that contact mitochondria. **(D)** Quantifications of contacts in control cells lacking Pex13-AID (shown in C) are based on $n$ = 3 experiments. Each color represents 1 experiment. Error bars represent standard error of the mean. $P$-values were calculated with a one-way ANOVA combined with a Tukey test. Underlying data for quantifications can be found in S1 Data.
(TIF)

**S5 Fig. Inhibition of peroxisomal protein import via an auxin inducible degron reduces PerMit contacts. (A and B)** Images are from indicated strains expressing Ant1-YFP (green), Tim50-CFP (blue), and Cat2-RFP-HA-PTS (red) photographed at indicated time points after addition of 2 mM indole-3-acetic acid (auxin). Arrows indicate peroxisomes that contact mitochondria. **(C)** Quantifications of contact in control cells lacking Pex13-AID (shown in B) are based on $n$ = 3 experiments. Each color represents 1 experiment. Error bars represent standard error of the mean. $P$-values were calculated with a one-way ANOVA combined with a Tukey test. Underlying data for quantifications can be found in S1 Data.
(TIFF)

**S6 Fig. PerMit contact are affected by depletion of Pex5. (A)** Fluorescence microscopic images of a galactose chase and shut-off experiment using a galactose-inducible conditional *pex5* mutant expressing RFP-PTS. Scale bars represents 5 μm. **(B)** Pex5 levels from indicated time points of a strain expressing Tim50-CFP, Ant1-YFP, and Ptc5ΔTM-RFP-PTS were analyzed by SDS-PAGE and immunoblot. Por1 served as loading control. **(C)** Quantifications of PerMit contacts in the course of the galactose chase and shut-off experiment. Note that this experiment was carried out via automated time-lapse imaging and does not directly reflect the Pex5 concentrations shown in S5B Fig, which were obtained from incubations in liquid media. Underlying data for quantifications can be found in S1 Data.
(TIFF)

**S7 Fig. Masking of the PTS1 of Lys12 does not affect growth under lysine-limiting conditions.** Serial dilutions of logarithmically growing cells of indicated strains were spotted on media with or without lysine and incubated at 30˚C.
(TIF)

**S8 Fig. Single-channel images of mutants identified in the high-content screening.** Subcellular localization of Ptc5-RFP-PTS (magenta) and Pex3-GFP (green) was analyzed in indicated strains using automated fluorescence microscopy. Shown are pictures from experiments

performed on putative hits. Δ*coq9* cells were used as a control as these show Ptc5-RFP-PTS localization in peroxisomes, but are affected in mitochondrial metabolism (Stehlik and colleagues). Δ*pex5* mutants show no peroxisomal targeting of the reporter Ptc5-RFP-PTS but only mitochondrial signal. Scale bars represents 5 μm.
(TIF)

**S9 Fig. Single-channel images related to Fig 6B.** Subcellular localization of Ptc5-RFP-PTS (red), the peroxisomal membrane protein Ant1-YFP (green), and the mitochondrial inner membrane protein Tim50-CFP (blue) in indicated strains was analyzed using fluorescence microscopy. Scale bar represents 5 μm.
(TIF)

**S10 Fig. Analysis of mitochondrial protein import in Δ*mdm10* cells and of Ptc5 sorting in different mitochondrial import or biogenesis mutants. (A)** Bona fide mitochondrial proteins C-terminally tagged with GFP at the endogenous locus in control and Δ*mdm10* cells were analyzed. Whole cell extracts were subjected to SDS-PAGE followed by immunoblot. Concentrations of protein extracts were adapted to each other to focus on processing. **(B)** Subcellular localization of Ptc5-RFP-PTS (red), the peroxisomal membrane protein Ant1-YFP (green), and the mitochondrial inner membrane protein Tim50-CFP (blue) in indicated strains was analyzed using fluorescence microscopy. **(C)** The truncated variant Ptc5$^{1-201}$-RFP lacking a PTS1 was expressed in mutants lacking components of the mitochondrial import machinery or the mitofusin Fzo1. Concentrations of protein extracts were adapted to each other to focus on processing. Whole cell extracts were subjected to SDS-PAGE followed by immunoblot. (D) Fluorescence microscopic pictures of Δ*pex11* cells expressing Ptc5-RFP-PTS (magenta) and Pex3-GFP (green). Scale bar represents 5 μm.
(TIF)

**S11 Fig. Lack of ERMES components changes the distribution of dual affinity proteins. (A and B)** Subcellular localization of indicated dual affinity proteins (red), the peroxisomal membrane protein Ant1-YFP (green), and the mitochondrial inner membrane protein Tim50-CFP (blue) in indicated strains was analyzed using fluorescence microscopy. Scale bars represent 5 μm.
(TIF)

**S12 Fig. Localization of RFP-PTS and Ptc5 derivatives in Δ*mdm10* cells. (A)** Localization of Ptc5-RFP-PTS (magenta, upper panel) and RFP-PTS (magenta, lower panel) in *mdm10* mutants expressing a synthetic ER–mitochondria tether (green) [43] and quantification of peroxisome number per cell in strains of the lower panel. **(B)** Fluorescence microscopic images showing expression of the peroxisomal marker RFP-PTS (magenta) in indicated strains. **(C)** Fluorescence microscopic images showing expression of Ptc5 derivatives in Δ*mdm10* cells (magenta). Quantifications show the ratio of peroxisomal versus total Ptc5-RFP-PTS signal. The peroxisomal membrane protein Ant1-YFP (green, B and C) and the mitochondrial protein Tim50-CFP (blue, C) served as marker proteins. Scale bar represents 5 μm. Quantifications are based on *n* = 3 experiments. Each color represents 1 experiment. Error bars represent standard error of the mean. *P*-values were calculated using a two-sided unpaired Student's *t* test. Underlying data for quantifications can be found in S1 Data.
(TIF)

**S13 Fig. Suppression of Δ*mdm12* does not prevent formation of aberrant peroxisomes. (A)** Serial dilutions of logarithmically growing cells of indicated strains were spotted on indicated media and incubated at 30˚C. **(B)** Fluorescence microscopic images of Δ*mdm10* cells (*SUP$^-$*) or

$\Delta mdm10$ cells with a suppressor mutation ($SUP^+$) co-expressing the peroxisomal marker Ant1-YFP (magenta) and the mitochondrial inner membrane protein Tim50-CFP (green). Scale bar represents 5 μm. **(C)** The number of peroxisomes per cell was quantified in the indicated strains. Quantifications are based on $n = 3$ experiments. Each color represents 1 experiment. Error bars represent standard error of the mean. *P*-values were calculated with a one-way ANOVA combined with a Tukey test. Underlying data for quantifications can be found in S1 Data. (TIF)

**S14 Fig. Expression of the synthetic PerMit tether in control cells.** Fluorescence microscopic images of cells co-expressing the peroxisomal marker RFP-PTS (red, A) or Pxp2-RFP-PTS (red, B), the peroxisomal membrane protein Ant1-YFP (green), and the mitochondrial protein Tim50-CFP. Scale bars represent 5 μm. (TIFF)

**S15 Fig. Analysis of peroxisomes in ERMES mutants upon expression of a synthetic tether.** Images of strains deleted for *mdm10* or *mdm34* co-expressing RFP-PTS (A, red) or Pxp2-RFP-PTS (B, red), the peroxisomal membrane protein Ant1-YFP (green), and the mitochondrial protein Tim50 (blue) either in the presence of Tom70-ProteinA-TA$_{Pex15}$ or a control protein. Scale bars represent 5 μm. (TIFF)

**S16 Fig. Genetic interactions of Mdm10. (A)** Indicated mutants expressing Pex3-GFP were analyzed by fluorescence microscopy (left). Quantifications of peroxisome number (right). **(B)** Indicated mutants expressing mCherry-PTS were analyzed by fluorescence microscopy (left). Quantifications of peroxisome number (right). Scale bars represent 5 μm. Quantifications are based on $n = 3$ experiments. Each color represents 1 experiment. Error bars represent standard error of the mean. *P*-values were calculated with a one-way ANOVA combined with a Tukey test. Underlying data for quantifications can be found in S1 Data. (TIF)

**S17 Fig. Single-channel images related to Fig 8E and 8G. (A)** Fluorescence microscopic images of indicated strains expressing mNeonGreen-Pex15 (green), mCherry-PTS (magenta), and Sec63-CFP (blue). **(B)** Fluorescence microscopic images indicated strains expressing mNeonGreen-Pex15 (green), mCherry-PTS (magenta), and Tim50-CFP (blue). **(C)** Fluorescence microscopic images of control and indicated mutant cells expressing mNeonGreen-Pex15 (green) and mCherry-PTS (magenta). Scale bar represents 5 μm. (TIF)

**S18 Fig. Suppression of the *get* deletion phenotypes by overexpression of GFP-Pex15g.** Fluorescence microscopic images of cells co-expressing GFP-Pex15g (green) under control of a galactose inducible promoter and RFP-PTS (red). Scale bars represent 10 μm. Upper panel: glucose grown cells; lower panel: 3 h galactose induction. Scale bar represents 5 μm. (TIF)

**S19 Fig. Overexpression of Pex15 derivatives induces contacts. (A)** Fluorescence microscopic images of cells co-expressing RFP-HDEL (magenta) PA-GFP-Pex15g (green) under control of the *MET25* promoter grown in the presence of 200 μm methionine. **(B)** Fluorescence microscopic images of indicated strains expressing mNeonGreen-Pex15 (green) under control of the *TEF1* promoter, mCherry-PTS (red), and Tim50-CFP (blue). **(C)** Induction of a chimeric RFP-Pex15-TA$_{Tom22}$ (red) via galactose addition in strains expressing Ant1-YFP (green) and Tim50-CFP (blue). Scale bar represents 5 μm. (TIF)

**S20 Fig. Synthetic genetic interaction between genes encoding protein involved in sorting and tethering. (A)** Fluorescence microscopic images of indicated strains expressing the mitochondrial marker mt-GFP (green) and the peroxisome marker mCherry-PTS (magenta). Scale bar represents 5 μm. **(B)** Differential centrifugation of post nuclear supernatants of indicated strains. Fractions were analyzed by SDS-PAGE and immunoblot. I, input; 13k, 13k × g pellet; 100k, 100k × g pellet; S, 100k × g supernatant. Pgk1 is a cytosolic protein, Kar2 an ER resident protein. **(C)** Growth assays of indicated mutants on indicated media. Shown are serial 10-fold dilutions starting with an $OD_{600}$ = 1.
(TIF)

**S1 Movie. Movement of peroxisomes in indicated strains.** Frames were captured with the following excitation times and gain settings: excitation time, 400 ms; gain, 228.
(MP4)

**S2 Movie. Movement of peroxisomes in indicated strains.** Frames were captured with the following excitation times and gain settings, control, 75 ms, gain 23; Δ*pex30*, 80 ms, gain 79; Δ*get2*, 100 ms, gain 63; Δ*mdm10*; 100 ms, gain 48.
(MP4)

**S1 Table. Candidates for proteins dually targeted to mitochondria and peroxisomes.**
(XLSX)

**S2 Table. Genes identified in the high-content screening.**
(XLSX)

**S3 Table. Quantification of Pxp2 at ERMES foci.**
(XLSX)

**S4 Table. Strains, plasmids, oligonucleotides.**
(XLSX)

**S1 Data. Data underlying quantification of imaging experiments.**
(XLSX)

**S1 Raw Images. Western blot data.**
(PDF)

## Acknowledgments

We thank Marisa Piscator for technical assistance in the Bölker laboratory. We acknowledge Bob Lesch for organizing data and Claudia Morales for support in the Schekman laboratory. We thank Amir Fadel and Lihi Gal for help with the automated library preparation and robotic screening procedures in the Schuldiner laboratory. We acknowledge Martin Thanbichler for sharing his microscope for time-lapse studies. We are grateful to Christof Taxis, Christian Renicke, Helle Ulrich, Jodi Nunnari, Michal Skruzny, Ralf Erdmann, and Roland Lill for antibodies, plasmids, and strains. We thank Uwe Maier and Björn Sandrock for comments on the manuscript.

## Author Contributions

**Conceptualization:** Thorsten Stehlik, Jason Lam, Gert Bange, Maya Schuldiner, Einat Zalckvar, Michael Bölker, Randy Schekman, Johannes Freitag.

**Formal analysis:** Elena Bittner, Thorsten Stehlik, Johannes Freitag.

**Funding acquisition:** Thorsten Stehlik, Gert Bange, Maya Schuldiner, Randy Schekman, Johannes Freitag.

**Investigation:** Elena Bittner, Thorsten Stehlik, Jason Lam, Lazar Dimitrov, Thomas Heimerl, Isabelle Schöck, Jannik Harberding, Anita Dornes, Nikola Heymons, Johannes Freitag.

**Methodology:** Elena Bittner, Jason Lam, Lazar Dimitrov, Thomas Heimerl, Anita Dornes, Einat Zalckvar, Johannes Freitag.

**Project administration:** Johannes Freitag.

**Resources:** Lazar Dimitrov, Thomas Heimerl, Isabelle Schöck, Nikola Heymons.

**Software:** Thorsten Stehlik.

**Supervision:** Gert Bange, Maya Schuldiner, Einat Zalckvar, Michael Bölker, Randy Schekman, Johannes Freitag.

**Validation:** Elena Bittner, Thorsten Stehlik.

**Visualization:** Thorsten Stehlik, Jannik Harberding.

**Writing – original draft:** Johannes Freitag.

**Writing – review & editing:** Elena Bittner, Thorsten Stehlik, Jason Lam, Gert Bange, Maya Schuldiner, Einat Zalckvar, Michael Bölker, Randy Schekman.

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
