## [Editor Report · Decision Letter 0]

20 Sep 2023

Dear Dr Freitag, 

Thank you for submitting the revised version of your manuscript entitled "Dual targeting regulates proximity between peroxisomes and partner organelles" for consideration as a Research Article by PLOS Biology. Also please accept my apologies for the delay - I thought I had sent the manuscript for review, but it didn't work because we first need you to complete again the metadata, as it is a new submission.

Please login to Editorial Manager where you will find the paper in the 'Submissions Needing Revisions' folder on your homepage. Please click 'Revise Submission' from the Action Links and complete all additional questions in the submission questionnaire.

Once your full submission is complete, your paper will undergo a series of checks in preparation for peer review. After your manuscript has passed the checks it will be sent out for review. To provide the metadata for your submission, please Login to Editorial Manager (https://www.editorialmanager.com/pbiology) within two working days, i.e. by Sep 22 2023 11:59PM.

Kind regards,

Ines

--

Ines Alvarez-Garcia, PhD

Senior Editor

PLOS Biology

---

## [Decision Letter · Decision Letter 1]

1 Dec 2023

Dear Dr Freitag,

Thank you for your patience while we considered your revised manuscript entitled "Dual targeting regulates proximity between peroxisomes and partner organelles" for publication as a Research Article at PLOS Biology. This revised version of your manuscript has been evaluated by the PLOS Biology editors, the Academic Editor and one of the original reviewers.

Based on the review, we are likely to accept this manuscript for publication, provided you satisfactorily address the remaining points raised by the reviewer and the Academic Editor (attached below). Please also make sure to address the data and other policy-related requests stated below. The Academic Editor also has a small comment that you will need to address.

In addition, we would like you to improve the title by expanding it and adding some details to make it more accessible to our broad audience. We would also like you to consider a suggestion to improve the title:

"Proteins that carry dual targeting signals can act as tethers between peroxisomes and partner organelles"

We expect to receive your revised manuscript within two weeks. 

*Published Peer Review History*

*Press*

Sincerely,

Ines

--

Ines Alvarez-Garcia, PhD

Senior Editor

PLOS Biology

Fig. 1B, D, E; Fig. 2C, E, G, H; Fig. 3C-D, F-I; Fig. 4C, D, F, G; Fig. 5D; Fig. 6B, G; Fig. 7A-C, G, F; Fig. 8C, F, H, I; Fig. 9D; Fig. S2A, B; Fig. S3C; Fig. S4D; Fig. S5C; Fig. S6C; Fig. S12A, C; Fig. S13C; Fig. S16A, B;

We require the original, uncropped and minimally adjusted images supporting all blot and gel results reported in an article's figures or Supporting Information files. We will require these files before a manuscript can be accepted so please prepare and upload them now. Please carefully read our guidelines for how to prepare and upload this data: https://journals.plos.org/plosbiology/s/figures#loc-blot-and-gel-reporting-requirements

Academic Editor's comments

Lines 340-342:

"Upon depletion of the previously identified PerMit tether Pex34, we visually observed more mitochondrial localization (Fig 5C and S8 Fig), however, we did not observe a difference in quantifications (Fig 5D), speaking for redundant modes of tethering."

This comment is very unclear: either there is a phenotype and it is quantifiable, or there isn't. It seems you are trying to say that you saw a phenotype, but the quantification says the opposite. Perhaps it would be better to just remove the comment.

Reviewers' comments

Rev. 2:

The authors have addressed some of my concerns, but did not analyze the functional role of the contacts and answered textually to some questions that I believed were to be addressed experimentally. I therefore suggest that the paper is further processed for publication, provided that the authors tone down some of the conclusions. For example, in the discussion I would avoid using the term tethers (because they did not demonstrate that these proteins act as tethers) and resort to the same wording used in the abstract. Moreover, the term "regularly" in the first paragraph of the discussion implies that this is the case in most instances, but this is far from being demonstrated. I would consider using the word "often" that is perhaps more appropriate. Please expand the discussion oh whether ERMES might affect proximity also between ER and mitochondri not as a physical tether, but because of its role in protein sorting and import. Refer also to the recent cryoEM structure by Wozny et al of ERMES PMID: 37165187

Other comments

1. Please correct the term "mitofusion 2" into Mitofusin 2 

2. when discussing the dual targeting of Mitofusin 2, please refer to the recent (2023) publication by Naon et al describing the mitofusin 2 splice variants that are targeted to the ER to tether it to mitochondria PMID: 37347868. 

3. introduction, line 58: "transfer of molecules", please add "and ions" as contact sites between ER-PM and ER-mitochondria are also crucially involved in the transfer of Ca2+

---

## [Editor Report · Decision Letter 2]

19 Jan 2024

Dear Dr Freitag,

Thank you for the submission of your revised Research Article entitled "Proteins that carry dual targeting signals can act as tethers between peroxisomes and partner organelles" for publication in PLOS Biology. On behalf of my colleagues and the Academic Editor, Sophie Martin, I am delighted to let you know that we can in principle accept your manuscript for publication, provided you address any remaining formatting and reporting issues. These will be detailed in an email you should receive within 2-3 business days from our colleagues in the journal operations team; no action is required from you until then. Please note that we will not be able to formally accept your manuscript and schedule it for publication until you have completed any requested changes.

PRESS

Sincerely, 

Ines

--

Ines Alvarez-Garcia, PhD

Senior Editor

PLOS Biology
